# Computing the Bias of Constant-step Stochastic Approximation with Markovian Noise

**Sebastian Allmeier**
Univ. Grenoble Alpes and Inria
F-38000 Grenoble, France.

**Nicolas Gast**
Univ. Grenoble Alpes and Inria
F-38000 Grenoble, France.

## Abstract

We study stochastic approximation algorithms with Markovian noise and constant step-size $\alpha$. We develop a method based on infinitesimal generator comparisons to study the bias of the algorithm, which is the expected difference between $\theta_n$ —the value at iteration $n$— and $\theta^*$ —the unique equilibrium of the corresponding ODE. We show that, under some smoothness conditions, this bias is of order $O(\alpha)$. Furthermore, we show that the time-averaged bias is equal to $\alpha V + O(\alpha^2)$, where $V$ is a constant characterized by a Lyapunov equation, showing that $\mathbb{E}\left[\bar{\theta}_n\right] \approx \theta^* + V\alpha + O(\alpha^2)$, where $\bar{\theta}_n = (1/n)\sum_{k=1}^n \theta_k$ is the Polyak-Ruppert average. We also show that $\bar{\theta}_n$ converges with high probability around $\theta^* + \alpha V$. We illustrate how to combine this with Richardson-Romberg extrapolation to derive an iterative scheme with a bias of order $O(\alpha^2)$.

## 1 Introduction

Stochastic approximation (SA) is a widely used algorithmic paradigm to solve fixed-point problems under noisy observations. While SA was introduced in the 1950s [30, 6], it is still widely used today in many applications to solve optimization problems, or to implement machine learning or reinforcement learning algorithms [23, 5]. A typical SA is a stochastic recurrence of the form

$$\theta_{n+1} = \theta_n + \alpha_n(f(\theta_n, X_n) + M_{n+1}), \tag{1}$$

where $\theta_n \in \mathbb{R}^d$ is a vector of parameters, $X_n$ and $M_{n+1}$ are sources of randomness, and $\alpha_n$ is the step-size. The step-size might be decreasing to $0$ as $n$ grow or can be fixed to a small constant (in which case we simply call it $\alpha$).

The goal of running a SA algorithm is to obtain a sequence $\theta_n$ that gets close to the root $\theta^*$ of the function $\bar{f}(\theta) = \mathbb{E}\left[f(\theta, X) + M \mid \theta\right]$, the expectation of the iterate (1), as $n$ goes to infinity. The SA procedure described in (1) models many algorithms that are used in machine learning, like first-order optimization and stochastic gradient descent [24, 16], and $Q$-learning or policy gradient algorithms [5, 35].

The theory of stochastic approximation has been studied extensively and is covered by a lot of textbooks, *e.g.*, [4, 8, 23]. This theory shows that the limit points of $\theta_n$ as $n$ grows are similar to the limit points of the ODE $\dot{x} = \bar{f}(x)$ [3, 9]. In particular, one can show that if all solutions of the ODE converge to $\theta^*$, then $\theta_n$ gets close to $\theta^*$ as $n$ grows large (under technical conditions on $f$, the step size $\alpha_n$, and the noise $X_n$ or $M_{n+1}$). The asymptotic behavior of $\theta_n$ depends on the nature of the step-size: When the step-size $\alpha_n$ depends on $n$ and converges to $0$ as $n$ goes to infinity, then $\theta_n$ generally converges almost surely to $\theta^*$. When $\alpha_n$ is kept constant and does not depend on $n$, then $\theta_n$ will in general *not* converge to $\theta^*$ but keeps oscillating around $\theta^*$, with fluctuation amplitudes of order $O(\sqrt{\alpha})$. In this work, we focus on the latter case and propose a method to quantify these fluctuations.

38th Conference on Neural Information Processing Systems (NeurIPS 2024).

We consider a SA algorithm with Markovian noise. That is, $\theta_n$ evolves as in (1) where the random variable $X_n$ evolves as a Markov chain that depends on $\theta_n$, and the random variable $M_{n+1}$ is a Martingale difference sequence. One thing that sets this paper apart from others is that we allow the Markovian noise to be dependent on $\theta_n$. This is particularly valuable in reinforcement learning application such as $Q$-learning where the exploration policy might depend on $\theta_n$ (this would for instance be the case for $\varepsilon$-greedy). It is known that when the step size $\alpha$ is constant, then the expectation of the iterates, $\mathbb{E}[\theta_n]$, does not converge to $\theta^*$ as $n$ goes to infinity but has a *bias* [41, 20]. The goal of our paper is to provide a framework to characterize and compute this bias.

Our main contribution is to provide a framework based on semi-groups, to quantify the convergence rate of $\theta_n$ to $\theta^*$. This framework is similar to Stein's method, that has been recently popularized to obtain accuracy results and refinement terms for fluid limits [10, 17, 18, 40]. It allows us to quantify the distance between the stochastic recurrence (1) and its deterministic counterpart (8) as a function of the distance between the infinitesimal generators of the stochastic and deterministic systems. By using this framework, we obtain two main results. Our first result (Theorem 1) states that, under smoothness conditions on $f$, there exists a constant $C > 0$ such that for all small enough $\alpha$:

$$\limsup_{n\to\infty} |\mathbb{E}[\theta_n] - \theta^*| \leqslant C\alpha \qquad \text{and} \qquad \limsup_{n\to\infty} |\mathbb{E}[(\theta_n - \theta^*)^2]| \leqslant C\alpha.$$

This guarantees that that the bias and variance of $\theta_n$ are of order *at most* $\alpha$.

A classical way to reduce the variance of SA is to use Polyak-Ruppert tail averaging [28, 31], *i.e.*, to look at the convergence of $\bar{\theta}_n = (1/n)\sum_{k=1}^n \theta_n$ instead of $\theta_n$. In this paper, we show that, for the constant step-size case, the use of Polyak-Ruppert averaging removes the variance (*i.e.* $\lim_{n\to\infty} \text{var}[\bar{\theta}_n] = 0$), but not the bias: Theorem 2 shows that there exists a vector $V$ and a constant $C' > 0$ such that:

$$\limsup_{n\to\infty} |\mathbb{E}[\bar{\theta}_n] - \theta^* - V\alpha| \leqslant C\alpha^2.$$

This shows that the bias of the averaging is *exactly* of order $\alpha$. We also show in Theorem 3 that $\bar{\theta}_n$ converges *in probability* to a point in $\theta^* + V\alpha + O(\alpha^2)$ as $n$ goes to infinity.

We also provide numerical simulation on a synthetic example that illustrates how to combine the Polyak-Ruppert averaging with a Richardson-Romberg extrapolation [19] to construct a stochastic approximation algorithm whose bias is of order $O(\alpha^2)$. This leads to an algorithm that enjoys both a small bias and a fast converge rate, compared to using a time-step of order $\alpha^2$.

*Roadmap.* The rest of the paper is organized as follows. We describe related work in Section 2. We introduce the model and give assumptions and notations in Section 3. We state the main results and provide numerical illustrations in Section 4. We give the main ingredients of the proofs in Section 5. The appendices contain technical lemmas and additional numerical examples.

## 2 Related work

Stochastic approximation algorithms were first introduced with decreasing step-size [30, 6]. Since these seminal papers, there has been a considerable amount of work aiming at characterizing the asymptotic properties of SA, by relating the asymptotic behavior of $\theta_n$ with the one of the ODE $\dot{\theta} = \bar{f}(\theta)$ [3, 9]. These topics are today well covered by textbooks such as [4, 8, 23].

In this original theory, the goal was to show that $\theta_n$ is close to the behavior of an ODE but not necessarily to obtain tight rates of convergence. This lead to the development of a new line of research focussing on non-asymptotic properties of SA [27]. This paper derives a unified framework to study the rate of convergence of stochastic gradient descent by relating it to SA with decreasing step-size, and martingale noise (*i.e.*, where the function $f$ does not depend on $X_k$ in (1)). This framework has been extended by a series of papers for applications in reinforcement learning, see for instance [11, 15, 29, 26], still in the decreasing step-size case, with the goal of understanding the fluctuations of the algorithm around the equilibrium.

For the decreasing step size, the fluctuations of $\theta_n$ around $\theta^*$ vanish as $n$ goes to infinity. This is not the case for constant step-size SA, for which understanding the magnitude of the fluctuations is therefore important. To bound the fluctuations, a part of the literature seeks to obtain bounds on the

mean squared error (MSE), $\mathbb{E}\left[(\theta_n - \theta^*)^2\right]$ as a function of the problem's parameter and the step-size $\alpha$. These papers show that the MSE is generally of order $O(\alpha)$, see for instance [33] for the case of linear model plus Markovian noise, or [13, 15, 14] for contractive stochastic approximations (with Markovian or martingale noise).

The case of constant-step size is also interesting because the $\theta_n$ has a bias that does not vanish as $n$ goes to infinity. In particular, the Polyak-Ruppert averaging does not converge to $\theta^*$: generally $\lim_{n\to\infty} \bar{\theta}_n \neq \theta^*$. It is shown in particular in [16, 20, 25, 32, 41, 42] that a constant step-size SA has an asymptotic bias of order $\alpha$, which shows that $\lim_{n\to\infty} \bar{\theta}_n = \theta^* + V\alpha + O(\alpha^2)$. Some of these papers, and in particular [16, 20], show that this bias characterization can be coupled with a Richardson-Romberg extrapolation to obtain a more accurate algorithm. In our paper, we also study the case of constant-step size SA with Markovian noise and use the same Polyak-Ruppert and Richardson-Romberg analysis. One of the distinguishable property of our model is that we allow the evolution of the Markovian noise $X_n$ to depend on $\theta_n$, whereas most of the cited paper consider $X_n$ has an external source of noise. This dependence is particularly interesting when studying asynchronous $Q$-learning algorithms [37, 36], for which the navigation policy is often derived from the current values of the parameter (a popular example is to use an $\epsilon$-greedy policy).

A second distinguishable feature of our paper is the methodology that we use. Our main tool is to relate the distance between the expectation of $\theta_n$ and $\theta^*$ as a distance between the infinitesimal generators of the stochastic recurrence (1) and of the deterministic recurrence (8). The method that we develop is tightly connected to Stein's method [34] and in particular with the line of work that use this method to obtain accuracy bounds for mean-field approximation [17, 22, 39, 40]. In particular, the start of our proof, which is to study a hybrid system composed of a stochastic and a deterministic recurrence in equation (9), can be seen as a discrete-time version of what is termed as a "classical trick" in [17, 22]. Apart from this difference between discrete and continuous time, one of the major features of our model is to have a Markovian noise. Our model can be viewed as a discrete-time version of the recent paper [1]. Some of our results (like our Theorem 1) are analogous to the results of [1] but others (like Theorem 2) are different and require time-averaging, mostly because of the possible periodicity of the discrete-time Markov chains. Also, the model of the current paper is slightly more general than the one of [1]. The fact that our methodology directly studies the expectation of $\theta_n$ makes it different from convergence results that use concentration equalities [12].

# 3 Model and preliminaries

## 3.1 Model and first assumptions

Throughout the paper $(\theta_n, X_n)_{n\geqslant 0}$ is a discrete-time stochastic process adapted to a filtration $(\mathcal{F}_n)_{n\geqslant 0}$. This stochastic process has two components of different nature. The first component $\theta$, that we call the parameter, is continuous and lives in a compact subset $\Theta \subset \mathbb{R}^d$: for each $n$: $\theta_n \in \Theta \subset \mathbb{R}^n$. The second component $X$ lives in a finite set $\mathcal{X}$, i.e., $X_n \in \mathcal{X}$.

The evolution of $\theta$ and $X$ are coupled in the following way. The parameter $\theta$ evolves according to the recurrence equation given by (1). At every time-step $n$, the process $X$ makes a Markovian transition according to a Markovian kernel $K(\theta_n)$. More precisely, for all $x, x' \in \mathcal{X}$ and $\theta \in \Theta$:

$$\mathbf{P}\left(X_{n+1} = x' \mid X_n = x, \theta_n = \theta, \mathcal{F}_n\right) = \mathbf{P}\left(X_{n+1} = x' \mid X_n = x, \theta_n = \theta\right) =: K_{x,x'}(\theta). \quad (2)$$

To obtain our results, we will make the following assumptions:

(A1) The process $M$ is a martingale difference sequence (that is: for all $n$: $\mathbb{E}\left[M_{n+1} \mid \mathcal{F}_n\right] = 0$) and its conditional covariance is $\mathbb{E}\left[M_{n+1}M_{n+1}^T \mid \mathcal{F}_n\right] := Q(\theta_n, X_n)$. Moreover, we assume that $\mathbb{E}\left[M_{n+1}M_{n+1}^T \mid \mathcal{F}_n \wedge X_{n+1}\right] = R(\theta_n, X_n, X_{n+1})$.

(A2) The functions $f$, $K$, $Q$ and $R$ are four times differentiable in $\theta$.

(A3) For any given $\theta \in \Theta$, the matrix $K(\theta)$ is unichain[1].

In addition to these three assumptions, we later add an assumption (A4) about the stability of the ODE around its fixed point. We do not state this assumption here as it needs extra definitions.

---

[1] We call a probability matrix unichain if the corresponding Markov chain has a single recurrent class and, possibly, some transient states.

## 3.2 Averaged values, average ODE and stability assumption

Assumption (A3) implies that a Markov chain with kernel $K(\theta)$ has a unique stationary measure $\pi(\theta)$, and we denote by $\pi_x(\theta)$ the stationary probability of $x \in \mathcal{X}$ for such a Markov chain. For any function $g$ defined on $\Theta \times \mathcal{X}$, we call $\bar{g}$ its *averaged* version, i.e., the function that associates to $\theta \in \Theta$ the value $\bar{g}(\theta)$, defined as:

$$\bar{g}(\theta) = \sum_{x \in \mathcal{X}} g(\theta, x) \pi_x(\theta). \tag{3}$$

It is shown in [1, Lemma 4], that under Assumption (A3), for each $\theta \in \Theta$, the kernel $K(\theta)$ has a unique stationary distribution $\pi(\theta)$. The value $\bar{g}(\theta)$ is equal to $\mathbb{E}\left[g(\theta, X)\right]$ where $X$ is distributed according to the stationary distribution associated to $K(\theta)$.

Following our notation introduced in (3), we denote by $\bar{f}$ the averaged version of $f$. By [1, Lemma 4], the function $\theta \mapsto \pi(\theta)$ is twice differentiable under assumptions (A2) and (A3), which implies that the function $\bar{f}$ is also twice differentiable. This implies, for an initial $\vartheta(0) = \theta$, the ODE $\dot{\vartheta} = \bar{f}(\vartheta)$ has a unique local solution, and we denote the value of this solution at time $t$ by $\phi_t(\theta)$. In order to prove this result, we will need that this ODE has a unique fixed point to which all trajectories converge and that this fixed point is an exponentially stable attractor. This is summarized in the following asusmption:

(A4) There exists a $\theta^*$ such that for any $\theta$, the solution of the ODE $\dot{\vartheta} = \bar{f}(\vartheta)$ is defined for all $t > 0$ and converges to $\theta^*$: $\lim_{t \to \infty} \phi_t(\theta) = \theta^*$ for all $\theta \in \Theta$. Moreover, the derivative of $\bar{f}$ at $\theta^*$ is Hurwitz (i.e., the real parts of all of its eigenvalues are negative).

By classical results on the stability of ODES, this assumption implies that the convergence to $\theta^*$ occurs exponentially fast, that is, there exists $a, b > 0$ such that for all $\theta \in \Theta$: $\|\phi_t(\theta) - \theta^*\| \leq a e^{-bt} \|\theta - \theta^*\|$. To prove that, one can use [21, Theorem 4.13], that shows that $\theta^*$ is an exponentially stable point of the ODE, *i.e.*, $\|\phi_t(\theta) - \theta^*\| \leq a' e^{-bt} \|\theta - \theta^*\|$ in a neighborhood $\mathcal{N}$ of $\theta^*$, because $D\bar{f}(\theta^* 0)$ is Hurwitz. Then, as $\Theta$ is compact, there exists $T$ such that $\phi_T(\theta) \in \mathcal{N}$ for all $\theta \in \Theta$. The result then follows by choosing $a = a' e^{Tb} \sup_{\theta \in \Theta} \|\theta - \theta^*\| / \sup_{\theta \in \mathcal{N}} \|\theta - \theta^*\|$.

## 3.3 Discussion on the assumptions and limits

Most of the assumptions used in the paper are classical when studying stochastic approximation algorithms with Markovian noise. In this section, we discuss the limits of each assumption (from the most classical one to the most original one). Assumption (A3) is necessary to define the notion of average dynamics $\bar{f}(\theta)$ and is therefore present in virtually all papers about Markovian noise stochastic approximation. The originality of our assumption is that we allow the transition kernel to depend on $\theta$. For assumption (A1), we add to the classical result the existence of a co-variance matrix $Q$, which is needed as it appears in the expression of $V$. Assumption (A4) imposes that the stochastic approximation has a unique exponentially stable attractor. Our results could probably be adapted to a model with multiple attractors (by using large deviation techniques similar to the one of [2, 38] for a two time-scale setting as ours) but this would be another paper.

One important assumption we make is that $\theta$ lives in a compact set $\Theta$. The bounded condition on $\Theta$ simplifies greatly the proofs because it allows us to use uniform bounds that do not depend on $\theta$ (for instance, the identity function $h(\theta) = \theta$ is bounded thanks to this assumption. Similarly, the time taken by the solution of the ODE $\phi_t(\theta)$ to reach $\theta^*$ is also uniformly bounded independently on $\theta$). This assumption could be relaxed by imposing high probability bounds (for instance, a large deviation result like [2] that would guarantee that $\theta_n$ stays close to $\theta^*$), or a bound on the higher-order moment or exponential moments of $\theta_n$. We left this result for future work as it would greatly impact the readability of the proofs.

The most questionable of our assumption is Assumption (A2) that imposes that all parameters of the problem are four times differentiable in $\theta$. While this assumption might seem as technical, the fact that the parameters are twice differentiable is crucial in our analysis. In particular, the constant $V$ does depend on the first two derivatives of the function $\bar{f}$. In general, if the parameters of the systems are not differentiable, then the bias will not be of order $O(\alpha)$ but of order $O(\sqrt{\alpha})$. Treating a non-differentiable $\bar{f}$ would need a completely different methodology. Our assumption of having

*four* times differentiable functions and not just twice has two reasons: it guarantees that the error $\|\bar{\theta}_n - (\theta^* + \alpha V)\|$ is $O(\alpha^2)$, and it simplifies the proof by allowing us to reuse Theorem 1 to obtain the bound $O(\alpha^2)$ in Theorem 2. We believe that if we only impose twice differentiability, this error would be $o(\alpha)$ (and in fact probably $\alpha^{3/2}$ as long as all derivatives are Lipschitz-continuous).

## 3.4 Notations

Recall that $\Theta$ is a compact subset of $\mathbb{R}^d$. We suppose that it is equipped with a norm $\|\cdot\|$. For a function $h : \mathcal{E} \to \mathbb{R}$, where $\mathcal{E} \subset \mathbb{R}^{d'}$, we denote by $\|h\| = \sup_{e \in \mathcal{E}} \|h(e)\|$ the supremum of this function. If $h$ is $i$ times differentiable, we denote by $D^i h$ its $i$th derivative. Its value evaluated in $e \in \mathcal{E}$ is denoted by $D^i h(e)$. It is a multi-linear map and we denote by $D^i h(e) a^{\otimes i}$ its value applied to $(a, \ldots, a)$ for a given $a \in \mathbb{R}^{d'}$. We denote by $\|D^i h\|$ the operator norm of its $i$th derivative, defined as $\|D^i h\| = \sup_{e \in \mathcal{E}, a \in \mathbb{R}^{d'}} \|D^i h(e) a^{\otimes j}\| / \|a\|^j$. By Taylor remainder theorem, for all $e \in \mathcal{E}$ and $a \in \mathbb{R}^{d'}$, we have: $\left\| h(e + a) - h(e) - \sum_{j=1}^{i-1} D^j h(e) a^{\otimes j} \right\| \leqslant \|D^i h\| \|a\|^i$.

We define by $\|D^{\leqslant i} h\| = \max(\|h\|, \max_{j \leqslant i} \|D^j h\|)$ the maximum of the norm of the first $i$th derivatives of $h$. We denote by $\mathcal{C}^i(\mathcal{E}, \mathbb{R})$ the set of functions whose first $i$ derivatives are bounded and by $\mathcal{C}^i_{\leqslant 1}(\mathcal{E}, \mathbb{R})$ the set of functions whose first $i$ derivatives are bounded by 1: $\|D^{\leqslant i} h\| \leqslant 1$. As $\mathcal{X}$ is a discrete set, the above notions extend to functions $h : \mathcal{E} \times \mathcal{X} \to \mathbb{R}$. For such a function, by abuse of notation, we call $D^i h$ the $i$th derivative with respect to the continuous variable only.

In the paper, the value at time $t$ of the solution of the ODE $\dot{\vartheta} = \bar{f}(\vartheta)$ starting in $\theta$ is denoted by $\phi_t(\theta)$. We will introduce later a quantity $\varphi_n(\theta)$ that corresponds to a Euler discretization of the ODE with constant step-size $\alpha$. To lighten notations, we will omit the dependence on $\alpha$ in the notation $\varphi$ but it should be clear that $\varphi$ depends on $\alpha$. To help distinguishing between the solutions of the the ODE $\phi_t$ and the discrete-time recurrence $\varphi_n$, we will reserve the index $t$ for a continuous time variable and the indices $n$, $k$ or $N$ for discrete-time indices.

To ease notations in some part of the proofs, we will sometimes use big-O notations, like $\mathcal{O}(1)$ or $\mathcal{O}(\alpha)$. When using this, we allow the hidden constants to depends on all parameters of the problems defined in Assumptions (A1) to (A4) but they cannot depend on varying quantities like an index $k$, $n$ or $\alpha$ or norm of functions, like $\|g\|$ or $\|h\|$ that are introduced in the proofs. Note that in all of our lemmas, we always consider functions whose norm is bounded by 1. This can, of course, be readily extended to functions not bounded by 1 by adding an extra factor like $\|D^{\leqslant i} h\|$. We avoid doing this to lighten the notations.

# 4 Main results and illustrations

This section presents the main theorems and illustrate their consequence. The proof of the theorems are postponed to the next Section 5.

## 4.1 Theoretical results

Our first result is Theorem 1 that shows that the expected value of $\theta_n$ is at distance at most $\mathcal{O}(\alpha)$ of the solution of the ODE. This shows that the bias of stochastic approximation is of order $\alpha$ with respect to the ODE.

---

**Theorem 1.** *Assume (A1)–(A4). Then, there exists a constant $C > 0$ and $\alpha_0$ such that for all $n$, $\alpha \leqslant \alpha_0$ and all $h \in \mathcal{C}^3_{\leqslant 1}(\Theta, \mathbb{R})$:*

$$|\mathbb{E}\left[h(\theta_n) - h(\phi_{\alpha n}(\theta_0))\right]| \leqslant \alpha C.$$

---

Note that the bound of Theorem 1 is valid independently of $n$. As we will see in the proof, this is a consequence of Assumption (A4). Without the latter assumption, one would naturally obtain a constant $C$ that grows exponentially with $n$. As $C$ does not depend on time, a direct consequence of Theorem 1 is to show that the asymptotic bias of $\theta_n$ is of order $\mathcal{O}(\alpha)$ as $n$ goes to infinity, that is:

$$\limsup_{n \to \infty} |\mathbb{E}\left[h(\theta_n)\right] - h(\theta^*)| \leqslant \alpha C. \tag{4}$$

Our second result is Theorem 2, that shows that the inequality of (4) is essentially an equality. This theorem provides an asymptotic expansion of the bias term in $\alpha$.

**Theorem 2.** *Assume (A1)–(A4). Then, there exists a constant $C' > 0$ and $\alpha_0$ such that for all $\alpha \leqslant \alpha_0$ and $h \in \mathcal{C}^5_{\leqslant 1}(\Theta, \mathbb{R})$:*

$$\limsup_{n \to \infty} \left| \frac{1}{n} \sum_{k=1}^{n} \mathbb{E}\left[ h(\theta_k) \right] - h(\theta^*) - \alpha V \right| \leqslant \alpha^2 C'. \tag{5}$$

An important difference between Theorem 1 and 2 is that the former studies the convergence of the iterates $\theta_n$ whereas the latter provides a refinement term for the average of the iterates, i.e., $\bar{\theta}_n := \frac{1}{n} \sum_{k=1}^{n} \mathbb{E}\left[ \theta_k \right]$, by showing that $\bar{\theta}_n$ is essentially equal to $h(\phi_\infty)$ plus a bias term $V\alpha$. One may wonder if Theorem 2 would be true for the (non-averaged) iterates $\theta_n$. The answer is no and a counter-example is provided in Appendix A. This example illustrates that when the Markovian component $X_n$ can be periodic, the $O(\alpha)$ term of $\mathbb{E}\left[ \theta_n \right]$ does not necessarily stabilize to a constant $V$ but can be periodic as well.

Theorem 2 concerns the convergence of the *expectation* of $\bar{\theta}_n$ but not the value of $\bar{\theta}_n$ itself. As we will see in Section 5, this is mostly due to our proof techniques that works with generators and therefore is most suitable to obtain precise convergence results for the expectation. In fact, we will prove in Section B.2 that we can obtain a high-probability convergence result as an almost-direct consequence of Theorem 2, as expressed by the following Theorem.

**Theorem 3.** *Assume (A1)–(A4). Then, there exists a constant $C'' > 0$ and $\alpha_0$ such that for all $\alpha \leqslant \alpha_0$:*

$$\lim_{n \to \infty} \mathbf{P}\left( \left| \bar{\theta}_n - (\theta^* + \alpha V) \right| \geqslant C'' \alpha^2 \right) = 0.$$

This result is illustrated on a synthetic example whose parameters are given in Appendix A.2, and for which $\theta^* = 1$. We run the stochastic approximation algorithm for various values of $\alpha$ and report the results in Figure 1. In all cases, we use the same source of randomness (the values of the Markov chain $X_n$ and of the the Martingale noise $M_{n+1}$ are the same for all trajectories). We plot a sample trajectory of $\theta_n$, $\bar{\theta}_n$, and $\bar{\theta}_{n/2:n}$ defined[2] as:

$$\bar{\theta}_{n/2:n} = \left\lceil \frac{n}{2} \right\rceil \sum_{k=\lfloor n/2 \rfloor + 1}^{n} \theta_k,$$

for $\alpha \in \{0.001/4, 0.001/8, 0.001/16\}$ (more values of $\alpha$ are displayed in Appendix A). We observe that if $\theta_n$ is quite noisy, the Polyak-Ruppert averaging $\bar{\theta}_n$ or $\bar{\theta}_{n/2:n}$ are much closer to $\theta^* = 1$, which is a clear advantage for $\bar{\theta}_{n/2:n}$ in terms of rate of convergence, especially for small values of $\alpha$.

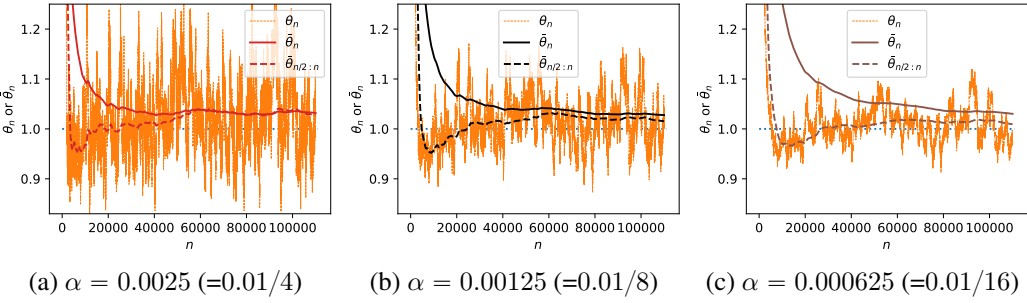

(a) $\alpha = 0.0025$ (=0.01/4)    (b) $\alpha = 0.00125$ (=0.01/8)    (c) $\alpha = 0.000625$ (=0.01/16)

Figure 1: Comparison of $\theta_n$, $\bar{\theta}_n$ and $\bar{\theta}_{n/2:n}$ for various $\alpha$.

## 4.2 The value of extrapolation: Illustration of Theorem 2 and 3

As we will see in the proof, the constant $V$ of Theorem 2 can be expressed as a function of the problem's parameters, which allows one to construct a quantity $\theta^* + V$ that is a tight approximation

---

[2]Note that if our theoretical results are presented for $\bar{\theta}_n$, they can be readily extended to $\bar{\theta}_{n/2:n}$.

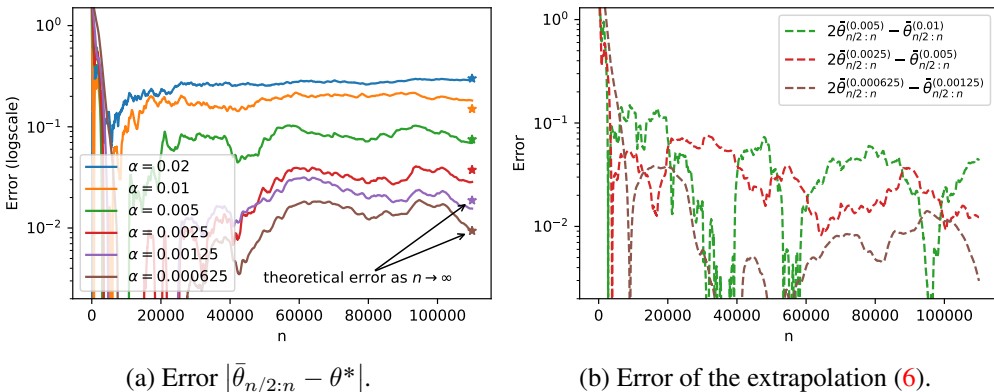

(a) Error $\left|\bar{\theta}_{n/2:n} - \theta^*\right|$.

(b) Error of the extrapolation (6).

Figure 2: Illustration of the error of $\bar{\theta}_n := \frac{1}{N} \sum_{k=1}^n \theta_k$ for various values of $\alpha = 0.02 \times 2^{-k}$ with $k \in \{0 \dots 5\}$ and of the error of the extrapolation (6) for $\alpha = 0.01$ and $\alpha = 0.005$.

of $\bar{\theta}_n$. Yet, stochastic approximation algorithms are most used when one does not have access to the problem's parameter. Here, we illustrate how we can run two algorithms with two different step-sizes in order to obtain an algorithm that has both a fast convergence rate and a high precision.

Let $\bar{\theta}_{n/2:n}^{(2\alpha)}$ and $\bar{\theta}_{n/2:n}^{(\alpha)}$ be two trajectories of the stochastic recurrence (1) each with respective step-size $2\alpha$ and $\alpha$. By Theorem 2, for large $n$ we have:

$$\bar{\theta}_{n/2:n}^{(2\alpha)} = \theta^* + 2V\alpha + \mathcal{O}(\alpha^2)$$

$$\bar{\theta}_{n/2:n}^{(\alpha)} = \theta^* + V\alpha + \mathcal{O}(\alpha^2)$$

We see that we can suppress the term in $2V\alpha$ and $V\alpha$ by using a linear combination of $\bar{\theta}_{n/2:n}^{(2\alpha)}$ and $\bar{\theta}_{n/2:n}^{(\alpha)}$, which leads to an equation that has a distance to $\theta^*$ that is of order $\mathcal{O}(\alpha^2)$:

$$2\bar{\theta}_{n/2:n}^{(\alpha)} - \bar{\theta}_{n/2:n}^{(2\alpha)} = \theta^* + \mathcal{O}(\alpha^2). \tag{6}$$

To explore the benefit of using the extrapolation (6), we plot in Figure 2 the error of the averaged iterate[3] $\left|\bar{\theta}_{n/2:n} - \theta^*\right|$, and the error of the extrapolation (6) for various values of $\alpha = 0.01 * 2^{-k}$, with $k = \{-1 \dots 4\}$. The parameters are the same as the one of Figure 1 and the various $\alpha$s use the same source of randomness. As we expect by the statement of the theorems, the error $\bar{\theta}_{n/2:n} - \theta^*$ is approximately equal to $V\alpha$ (the stars on the right are the theoretical values of $\lim_{n\to\infty} \left|\bar{\theta}_{n/2:n} - \theta^*\right|$). On the right panel, we observe that $2\bar{\theta}_n^{(\alpha)} - \bar{\theta}_n^{(2\alpha)}$ is closer to $\theta^*$ than the corresponding $\bar{\theta}_n^{(\alpha)}$ (the scale of the $y$-axis is the same for both panels). Note that even for $n = 10^5$, the valus of $\alpha$ are still noisy.

# 5 Proof overview and generator method

## 5.1 Proof overview

The first idea of our proof is to construct new random variables $\varphi_k(\theta_{n-k})$ that corresponds to a system where we apply the stochastic recurrence (1) for $k$ steps and then apply the deterministic recurrence (8) up to time $n$. By introducing these new variables and comparing their expectation for $k$ and $k + 1$, we reduce our problem to a comparison of the generators of the stochastic and of the deterministic system. This step is done in Section 5.2. In an independent lemma, we use Assumption (A4) to show that the derivatives of the function $\varphi_n$ converge exponentially fast to 0 as $n$ goes to infinity. This leads to Lemma 6 whose proof is technical and postponed to Section C.

Once these two basic points are obtained, we will use them to prove Proposition 4 which shows that the expectation of $\theta_n$ is close to the solution of the deterministic ODE. More precisely, this

---

[3]We only plot the error of $\bar{\theta}_{n/2:n}$ because it has the same limit as $\bar{\theta}_n$ but converges faster.

proposition shows that there exists a bounded sequence of constants $V_n^{(\alpha)}$ and a constant $C > 0$ such that for all $n \in \mathbb{N}$: $\left| \mathbb{E}\left[ h(\theta_n) - h(\phi_{n\alpha}(\theta_0)) \right] - \alpha V_n^{(\alpha)} \right| \leqslant C\alpha^2$. A main difficulty is to show that $V_n^{(\alpha)}$ is bounded. For that we will use Lemmas 7 and 8 that use properties of the Poisson Equation (21) to show that if the functions $g_k$ do not vary too much with $k$, then they can be replaced by their averaged versions:

$$\left| \sum_{k=0}^{n-1} g_k(\theta_k, X_k) - \sum_{k=0}^{n-1} \mathbb{E}\left[ \bar{g}_k(\theta_k) \right] \right| \leqslant C\left(1 + \alpha \sum_{k=0}^{n-1} \|g_{k+1} - g_k\|\right). \tag{7}$$

Theorem 1 is a direct consequence of Proposition 4. by using that $V_n^{(\alpha)}$ are bounded.

The next step is to show that the time average versions of $V_n^{(\alpha)}$, $\overline{V}_n^{(\alpha)} = n^{-1} \sum_{k=1}^{n} V_n^{(\alpha)}$, converges (as $n$ goes to infinity) to a term $V^{(\alpha)}$. Here, the main technical difficulty is to obtain a refinement of the averaging property of (7), which is done in Lemmas 9 and 10. We then use Theorem 1 and Lemma 11 to show that $V^{(\alpha)} \approx V + \mathcal{O}(\alpha)$ where $V$ is given in Proposition 5. This gives Theorem 2. As a side-product, Lemma 11 also shows that the constant $V$ can be computed by solving a linear system. The high-probability bound (Theorem 3) is a consequence of Theorem 2.

The proof structure is illustrated in Figure 3 that shows the dependencies between the lemmas.

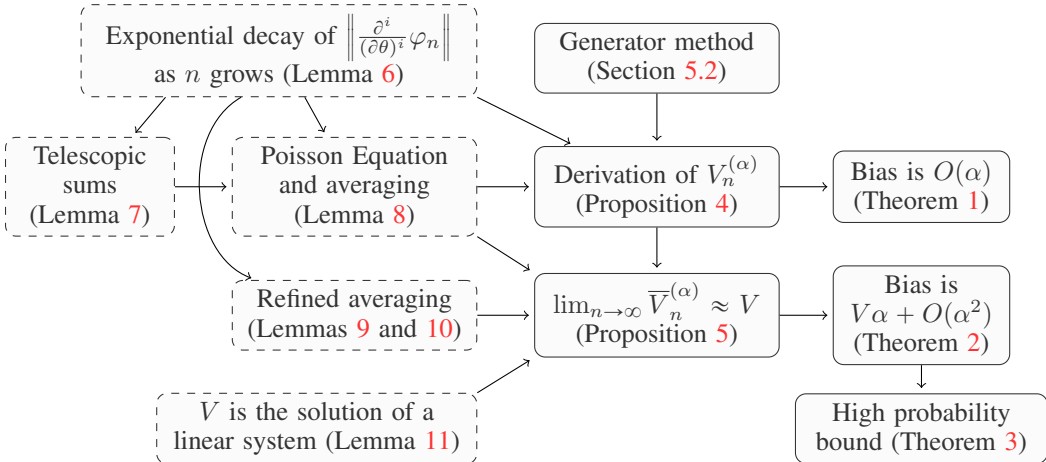

Figure 3: Overview of the proof. The dashed rectangles indicate the lemmas that are proven in Appendix C.

## 5.2 Deterministic recurrence and comparison of generators

To compare the stochastic variable $\theta_n$ and the solution of the ODE $\dot{\vartheta} = \bar{f}(\vartheta)$, we introduce a deterministic recurrence equation, that is a first-order discretization of the ODE. This recurrence equation is obtained by replacing the sources of randomness of (1) by their expectation. The stochastic approximation (1) contains two sources of randomness: $M_{n+1}$ and $X_n$. In our analysis, we will use a deterministic counterpart of (1) that corresponds to setting the noise $M_{n+1}$ to 0 and to using $\bar{f}(\theta)$ instead of $f(\theta_n, X_n)$. More precisely, for an initial value $\theta \in \Theta$ and $k \in \mathbb{Z}^+$, we define $\varphi_n(\theta)$ as:

$$\varphi_{n+1}(\theta) = \varphi_n(\theta) + \alpha\bar{f}(\varphi_n(\theta)), \tag{8}$$

with the convention that $\varphi_0(\theta) = \theta$.

Let $h : \Theta \to \mathbb{R}$ be an arbitrary function. By using the definition of the deterministic recurrence, for any $k \in \{0 \dots n\}$, we introduce the variable

$$z_{n,k} := \mathbb{E}\left[ h(\varphi_{n-k}(\theta_k)) \right]. \tag{9}$$

The quantity $z_{n,k}$ is the expected value of a recurrence at time $n$ if one starts by applying the stochastic recurrence for the first $k$ steps and then the deterministic recurrence for the remaining $n-k$

steps. Our proof method consists in obtaining precise bounds on the difference $\mathbb{E}\left[\theta_n\right] - \mathbb{E}\left[\varphi_n(\theta_0)\right]$. By using the notation $z_{n,k}$, this quantity is equal to $z_{n,n} - z_{n,0}$. To obtain a bound on this quantity, we will use a trick, that essentially consists in comparing $z_{n,k+1} - z_{n,k}$. This quantity is simpler to analyze because the only modification between the two is to replace one stochastic transition by one deterministic transition. We can then recover the original bound by using that:

$$\mathbb{E}\left[\theta_n\right] - \varphi_n(\theta_0) = z_{n,n} - z_{n,0} = \sum_{k=0}^{n-1} z_{n,k+1} - z_{n,k}. \tag{10}$$

This method is illustrated in Figure 4 on a synthetic example whose parameters are given in Appendix A.3. In the right panel, plot two functions (for two different values of $k$) that are equal to $\theta_n$ for $k \leqslant n$ and to $\varphi_{n-k}(\theta_k)$ for $k > n$.

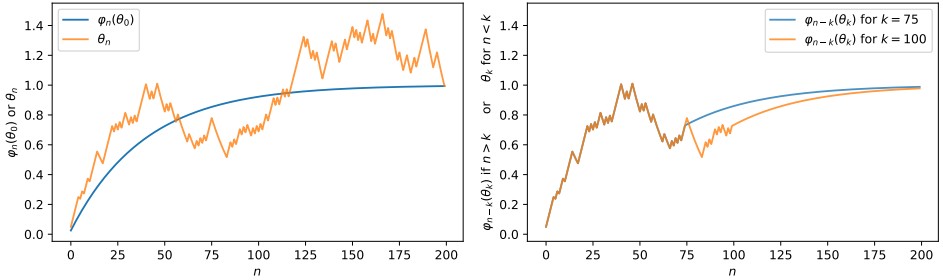

(a) Stochastic vs deterministic recurrences     (b) Impact of changing $k$ in $\varphi_{n-k}(\theta_k)$

Figure 4: Illustration of the behavior of $\theta_n$, $\varphi_n(\theta_0)$ and $\varphi_{n-k}(\theta_k)$.

## 5.3 Derivation of $V_n^{(\alpha)}$ and of $V$ by comparing the generators

Using the above notations, we are now ready to prove the first proposition.

**Proposition 4.** *Assume (A1)–(A4). Then, there exists a constant $C > 0$ and $\alpha_0$ such that for all $n$, $\alpha \leqslant \alpha_0$ and all $h \in \mathcal{C}_{\leqslant 1}^3(\Theta, \mathbb{R})$:*

$$\left|\mathbb{E}\left[h(\theta_n) - h(\phi_{n\alpha}(\theta_0))\right] - \alpha V_n^{(\alpha)}\right| \leqslant C\alpha^2,$$

*where the quantity $V_n^{(\alpha)}$ is given by:*

$$V_n^{(\alpha)} := \sum_{k=0}^{n-1} \mathbb{E}\bigg[ D(h \circ \varphi_{n-(k+1)})(\theta_k)(f(\theta_k, X_k) - \bar{f}(X_k))$$

$$- D^2(h \circ \varphi_{n-(k+1)})(\theta_k)\left((f(\theta_k, X_k) - \bar{f}(X_k))^{\otimes 2} + Q(\theta_k, X_k)\right)\bigg].$$

*Moreover, $V_n^{(\alpha)}$ is bounded independently of $\alpha$ and $n$.*

*Proof.* Following the definition of the stochastic and deterministic recurrences, the difference $z_{n,k} - z_{n,k+1}$ that appears in (10) is equal to

$$z_{n,k+1} - z_{n,k} = \mathbb{E}\left[h(\varphi_{n-k}(\theta_k)) - h(\varphi_{n-(k+1)}(\theta_{k+1}))\right]$$

$$= \mathbb{E}\big[h \circ \varphi_{n-(k+1)}(\theta_k + \alpha\bar{f}(\theta_k))$$

$$- h \circ \varphi_{n-(k+1)}(\theta_k + \alpha(f(\theta_k, X_k) + M_{k+1}))\big] \tag{11}$$

If $g : \mathbb{R}^d \to \mathbb{R}$ is a thrice differentiable function whose third derivative is bounded by $\|D^3 g\|$, by using a Taylor expansion, we have that $g(x+a) - g(x+b) = Dg(x)(a-b) + D^2g(x)(a-b)^{\otimes 2} + R$, where $R$ is a remainder term that is smaller than $\frac{c}{6}\|a-b\|^3$. In our proof, we use this Taylor

expansion with the function $g = h \circ \varphi_{n-(k+1)}$, $x = \theta_k$, $a = \alpha(f(\theta_k, X_k) + M_{k+1})$ and $b = \alpha \bar{f}(\theta_k)$. Applying this to (11) shows that

$$z_{n,k+1} - z_{n,k} = \alpha A_{n,k} + \frac{\alpha^2}{2} B_{n,k} + R_{n,k}, \tag{12}$$

where $A_{n,k}$ and $B_{n,k}$ are equal to

$$A_{n,k} := \mathbb{E}\left[D(h \circ \varphi_{n-(k+1)})(\theta_k)(f(\theta_k, X_k) + M_{k+1} - \bar{f}(X_k))\right]$$
$$B_{n,k} := \mathbb{E}\left[D^2(h \circ \varphi_{n-(k+1)})(\theta_k)(f(\theta_k, X_k) + M_{k+1} - \bar{f}(X_k))^{\otimes 2}\right],$$

and $R_{n,k}$ is a remainder term.

By assumption (A1), the conditional expectation and variance of the martingale term are equal to $\mathbb{E}\left[M_{k+1} \mid \mathcal{F}_k\right] = 0$ and $\mathbb{E}\left[M_{k+1}^{\otimes 2} \mid \mathcal{F}_k\right] = Q(\theta_k, X_k)$. By using the total law of expectation, this shows that $A_{n,k}$ and $B_{n,k}$ can be simplified as:

$$A_{n,k} := \mathbb{E}\left[D(h \circ \varphi_{n-(k+1)})(\theta_k)(f(\theta_k, X_k) - \bar{f}(X_k))\right]$$
$$B_{n,k} := \mathbb{E}\left[D^2(h \circ \varphi_{n-(k+1)})(\theta_k)\left((f(\theta_k, X_k) - \bar{f}(X_k))^{\otimes 2} + Q(\theta_k, X_k)\right)\right].$$

By definition, $V_n^{(\alpha)} = \sum_{k=0}^{n-1}(A_{n,k} + \alpha B_{n,k}/2)$. Hence, combining this with (10) shows that

$$\mathbb{E}\left[h(\theta_n) - h(\phi_{n\alpha}(\theta_0))\right] = V_n^{(\alpha)} + \sum_{k=0}^{n-1} R_{n,k}.$$

To complete the proof, it remains to be shown that there exists a constant such that $\sum_{k=0}^{n-1} R_{n,k} \leqslant C\alpha^2$ and that $V_n^{(\alpha)}$ is bounded independently of $\alpha$ and $n$. There are three terms, corresponding to $R$, $B$, and $\sum_{k=1}^{n} A_{n,k}$:

**Case of** $R$. In Lemma 6, we show that the norm of the derivatives of $h \circ \varphi_{n-k}$ are smaller than $c_1 e^{-c_2\alpha(n-k)}$. As the derivatives of $h$ are bounded by 1, this implies there exists $c$ independent of $\alpha$ and $n$ such that $\left\|D^3(h \circ \varphi_{n-k})\right\| \leqslant c e^{-c_2\alpha(n-k)}$. Hence, there exists $c' > 0$ such that:

$$\sum_{k=0}^{n-1} \|R_{n,k}\| \leqslant c'\alpha^3 \sum_{k=0}^{n-1} e^{-c_2\alpha(n-k)} \leqslant c'\alpha^3 \sum_{k=0}^{\infty} e^{-c_2\alpha k} \leqslant c'c_3\alpha^2.$$

**Case of** $B$. The proof that $\sum_{k=1}^{n} \alpha B_{n,k}$ is bounded independently of $n$ and $\alpha$ is similar since the factor $\alpha$ cancels out with the factor $1/\alpha$ that comes out of $\sum_{k=0}^{\infty} e^{-c_2\alpha k}$.

**Case of** $A$ Showing that $\sum_{k=1}^{n} A_{n,k}$ is bounded independently of $n$ and $\alpha$ is more subtle and uses the fact that $\bar{f}$ is the "averaged" version of $f$ with respect to the transition kernel $K$. The proof of this is a consequence of the first point of Lemma 8 that is stated in Section C. □

The proof of the main results is then a consequence of the next proposition, whose proof is given in Appendix B. This proposition shows that the Cesaro limit of $V_n^{(\alpha)}$ is approximately equal to a term $V$ that does not depend on $\alpha$ plus a term of order $O(\alpha)$.

> **Proposition 5.** *Assume (A1)–(A4). Then, there exist a constant $C > 0$ and $\alpha_0 > 0$ such that for all $h \in \mathcal{C}_{\leqslant 1}^5(\Theta, \mathbb{R})$, there exists a vector $V$ such that for all $n$, $\alpha \leqslant \alpha_0$:*
>
> $$\left|\overline{V}_n^{(\alpha)} - V\right| \leqslant C(\alpha + \frac{1}{n\alpha}),$$
>
> *where $\overline{V}_n^{(\alpha)} := n^{-1}\sum_{k=1}^{n} V_k^{(\alpha)}$ with $V_k^{(\alpha)}$ as in Proposition 4.*

# 6 Conclusion / discussion

In this paper, we presented an analysis of the bias of constant-step size algorithms with Markov noise. We developed a novel technic, based on generator comparison, that allows to obtain a very fine comparison between the expectation of the stochastic trajectory and the value of its deterministic counterpart. This methodology is quite generic and we believe that it could easily be adapted to obtain more general results. We are in particularly targeting: relaxing the bounded of $\Theta$, and obtaining non-asymptotic results.

# 7   Acknowledgements

The authors would like to thank the three anonymous reviewers for their insightful comments about the paper. This work was supported by the ANR (Agence National de la Recherche), via the project REFINO (ANR-19-CE23-0015).

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

# Appendix

## A  Additional numerical results

### A.1  Reproducibility

All parameters given in this section should suffice to reproduce all figures. The total computation time to obtain all figures of the paper does not exceeds a few minutes (on a 2018 laptop, all implementation being done in Python/Numpy/Matplotlib). To ensure reproducibility, the code necessary to reproduce the paper (including latex, code to run the simulations and code to plot the figures) is available at `https://github.com/ngast/paper_bias_stochastic_approximation2024`.

### A.2  Parameters for the example of Figure 1 and Figure 2

For Figure 1 and 2, the state space of the Markovian part is $\mathcal{X} = \{0, 1\}$, and the transition matrix is

$$K(\theta) = \begin{pmatrix} \sin(\theta)^2 & \cos(\theta)^2 \\ \cos(\theta)^2 & \sin(\theta)^2 \end{pmatrix}.$$

The drift is $f(\theta, x) = \theta - 2x$. The martingale noise $M_{n+1}$ is a sequence of *i.i.d.* random variables with $\mathbf{P}\left(M_{n+1} = 1 \mid \mathcal{F}_n\right) = \mathbf{P}\left(M_{n+1} = -1 \mid \mathcal{F}_n\right) = 0.5$. One can show that $\theta^* = 1$. The initial value of the algorithm is set to $\theta_0 = 3, X_0 = 0$.

In Figure 5, we complete the results of Figure 1 by showing more values of $\alpha$. We observe that in all cases, the averaged values $\bar{\theta}_n$ and $\bar{\theta}_{n/2:n}$ are much close to $\theta^* = 1$ than the original $\theta_n$ that is very noisy. This explain why using a Polyak-Ruppert averaging is important. Also, if for large values of $\alpha \in \{0.02, 0.01, 0.005\}$, the two quantities $\bar{\theta}_n$ and $\bar{\theta}_{n/2:n}$ have similar behavior, the situation is quite different when $\alpha$ gets smaller. In the latter case, the convergence of $\bar{\theta}_{n/2:n}$ is much faster than the one of $\bar{\theta}_n$ because of the transient behavior of $\theta_n$ at the beginning. This is why we use $\bar{\theta}_{n/2:n}$ in our simulations.

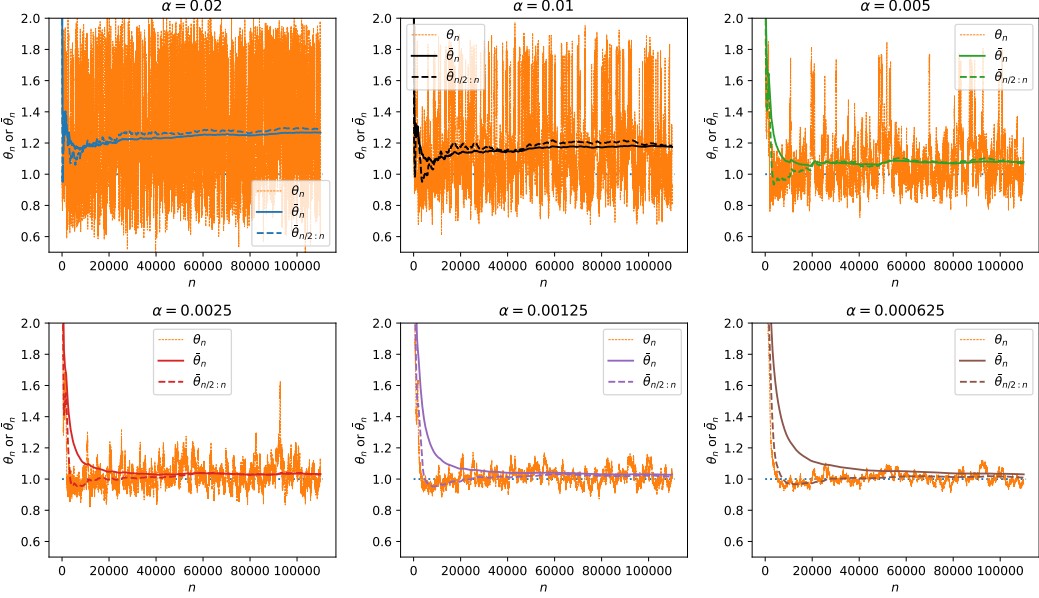

Figure 5: Behavior of $\theta_n$, $\bar{\theta}_n$ and $\bar{\theta}_{n/2:n}$ for various values of $\alpha$. All $y$-axis have the same scale.

### A.3 Parameters for Figure 4

For Figure 4, the state space of the Markovian part is $\mathcal{X} = \{0, 1\}$ and the transition matrix is

$$K(\theta) = \begin{pmatrix} 0.5 & 0.5 \\ 0.5 & 0.5 \end{pmatrix}.$$

The drift is $f(\theta, x) = 1 - (1 + \theta)x$. The initial state is $\theta_0 = X_0 = 0$ and there is no martingale noise[4]. The step-size is set to $\alpha = 0.05$.

### A.4 Why is Theorem 1 for $\theta_n$ and Theorem 2 and 3 for $\bar{\theta}_n$?

Note that compared to (4) of Theorem 1, the result of (5) in Theorem 2 is stated for the averaged iterates $\frac{1}{N} \sum_{n=1}^{N} \mathbb{E}[\theta_n]$ and not directly for $\theta_n$. In fact, there are many cases for which (5) is also valid for $\theta_n$. In particular, if $\mathbb{E}[h(\theta_n)]$ converges as $n$ goes to infinity, then one necessarily has:

$$\limsup_{n \to \infty} |\mathbb{E}[h(\theta_n)] - h(\theta^*) - \alpha V| \leqslant \alpha^2 C'.$$

This is for instance the case if $(\theta_n, X_n)$ is an ergodic Markov chain. Yet, there are cases for which the result of (5) only holds for the averaged iterates. This occurs typically when $X_n$ has a periodic behavior, in which case the averaged iterates eliminate eventual oscillations caused by periodic noise sequences. To illustrate this, we consider a one-dimensional model with $f(\theta, x) = 2x - \theta$, $X_{n+1} = 1 - X_n$ and $M_{n+1} = 0$. This model satisfies all assumptions of the problems and $\theta^* = 1$. The initial conditions are set to $\theta_0 = 0.85$, and $X_0 = 0$. The system is therefore deterministic. In Figure 6 we plot the trajectories of $\theta_n$ and of $\bar{\theta}_N^{(\alpha)} := \frac{1}{N} \sum_{n=1}^{N} \theta_n$ for two stepsize values: $\alpha = 0.05$ and $\alpha = 0.1$. We observe that, as stated by Theorem 1, the smaller is $\alpha$, the closer is $\lim_{n \to \infty} \theta_n$ from $\theta^*$. One can also observe that $(\theta_n - 1)/\alpha$ oscillates and does not converge as $n$ goes to infinity, contrary to $(\bar{\theta}_n - 1)/\alpha$ that does converge (to 0 here).

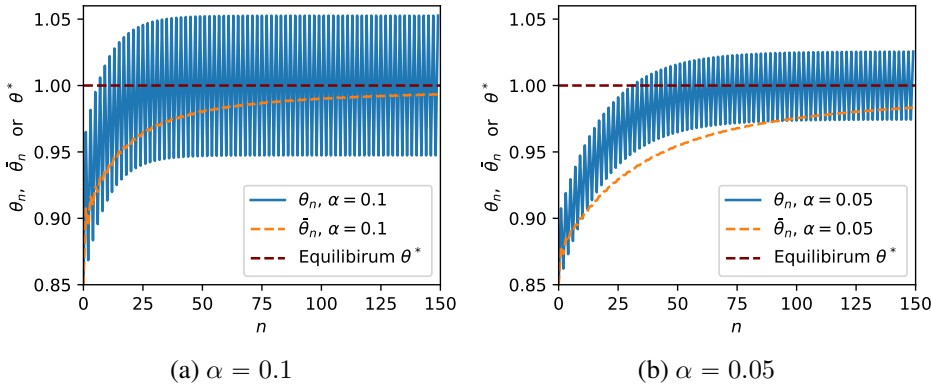

(a) $\alpha = 0.1$          (b) $\alpha = 0.05$

Figure 6: Example of a model with a periodic $X_n$ illustrating the necessity of averaging the iterates.

## B Proof of the main results

### B.1 Proof of Proposition 5

In this section, we prove Proposition 5 that states, that under (A1)–(A4), there exist a constant $C > 0$ and $\alpha_0 > 0$ such that for all $h \in \mathcal{C}_{\leqslant 1}^5(\Theta, \mathbb{R})$, there exists a vector $V$ such that for all $n, \alpha \leqslant \alpha_0$:

$$\left| \overline{V}_n^{(\alpha)} - V \right| \leqslant C(\alpha + \frac{1}{n\alpha}),$$

where $\overline{V}_n^{(\alpha)} := n^{-1} \sum_{k=1}^{n} V_k^{(\alpha)}$ with $V_k^{(\alpha)}$ as in Proposition 4.

---

[4]Note that because of the form of the matrix $K(\theta)$, the process $X_n$ is memoryless. Hence, one could build the same model with a martingale noise instead of the variable $X$.

This proposition shows that the Cesaro limit of the constants $V_n^{(\alpha)}$ defined in Proposition 4 converges to a limit that is equal to $V + \mathcal{O}(\alpha)$. In particular, this result implies that

$$\mathbb{E}\left[\frac{1}{k}\sum_{k=1}^n h(\theta_n)\right] = h(\theta^*) + \alpha V + \mathcal{O}(\alpha^2 + \frac{1}{n\alpha}).$$

*Proof of Proposition 5.* Recall from the proof of Proposition 4 that $V_n^{(\alpha)} = \sum_{k=0}^{n-1}\mathbb{E}\left[A_{n,k} + \alpha B_{n,k}/2\right]$. We first treat the term $\sum_{k=0}^{n-1} B_{n,k}$ that is the easiest (essentially because it is multiplied by a factor $\alpha$). Applying Lemma 8 to the function $g(\theta, X) := (f(\theta, X) - \bar{f}(X))^{\otimes 2} + Q(\theta, X)$ shows that:

$$\sum_{k=0}^{n-1}\mathbb{E}\left[B_{n,k}\right] = \sum_{k=0}^{n-1}\mathbb{E}\left[D^2(h\circ\varphi_{n-(k+1)})(\theta_k)g(\theta_k, X_k)\right]$$

$$= \sum_{k=0}^{n-1}\mathbb{E}\left[D^2(h\circ\varphi_{n-(k+1))})(\theta_k)\bar{g}(\theta_k)\right] + \mathcal{O}(1).$$

By applying Theorem 1, this shows that

$$\lim_{n\to\infty}\sum_{k=0}^{n-1}\mathbb{E}\left[B_{n,k}\right] = \sum_{k=0}^\infty D^2(h\circ\varphi_k)(\theta^*)\bar{g}(\theta^*) + \mathcal{O}(1), \tag{13}$$

where $\bar{g}(\theta) = \sum_{x\in\mathcal{X}}\pi_x(\theta)\left((f(\theta, x) - \bar{f}(x))^{\otimes 2} + Q(\theta, x)\right)$.

The treatment of $A_{n,k}$ requires more work, and in particular one can show that $\sum_{k=0}^{n-1} A_{n,k}$ does not necessarily converge as $n$ goes to infinity[5]. This is why we study the convergence of $\bar{A}_N = \sum_{n=1}^N\sum_{k=0}^{n-1} A_{n,k}$ and not the one of $A_{n,k}$. By applying Lemma 10 with $g = f$ and then applying Theorem 1, it holds that $\lim_{N\to\infty}\left|\bar{A}_N + \alpha m(\theta^*)\right| \leqslant C\alpha$, with $m(\theta^*)$ equal to

$$m(\theta^*) = \sum_{x,x'\in\mathcal{X}}\sum_{n=0}^\infty D(D(h\circ\phi_n)(\theta^*)G_f(\theta^*, x'))(f(\theta^*, x) + R(\theta^*, x, x'))K_{x,x'}(\theta^*)\pi_x(\theta^*)$$

Note that to apply Theorem 1, we need $m$ to be three times differentiable which is why we need $h$ to be four times differentiable.

Let $V^{(\alpha)} = \alpha(\sum_{k=0}^\infty D^2(h\circ\varphi_k)(\theta^*)\bar{g}(\theta^*) + m(\theta^*))$. By using Lemma 11, there exists a vector $V$ such that $\left\|V^{(\alpha)} - V\right\| \leqslant C\alpha$. $\qquad\square$

## B.2 Proof of Theorem 3 (High-probability bound)

Fix some $i \in \{1\ldots d\}$ and let $Y_n := (\theta_n - \theta^* - \alpha V)_i$, be the $i$th component of the vector where the constant $V$ is the one defined in Proposition 5 when applied to the function $h(\theta) = \theta - \theta^*$. By Proposition 4 and Proposition 5, there exists $C > 0$ such that for $n$ large enough:

$$\frac{1}{n}\sum_{k=1}^n\mathbb{E}\left[Y_k\right] \leqslant C(\alpha^2 + \frac{1}{\alpha n})$$

It is easy to modify the proofs of those propositions (by adding a conditioning on $\mathcal{F}_{mK}$ and starting the summation at $mK + 1$) to show that for all $K$ and for $m$ large enough:

$$\frac{1}{K}\sum_{k=mK+1}^{(m+1)K}\mathbb{E}\left[Y_k \mid \mathcal{F}_{mK}\right] \leqslant C(\alpha^2 + \frac{1}{\alpha K}).$$

Let $Z_m = \frac{1}{K}\sum_{k=mK+1}^{(m+1)K} Y_m$. The above result shows that $\sum_{m=1}^M Z_m - CM(\alpha^2 + \frac{1}{\alpha K})$ is a supermartingale. Moreover, as $\Theta$ is bounded, there exists $C' > 0$ (independent of $C$ and $\alpha$) such that $|Z_n - C(\alpha^2 + \frac{1}{\alpha K})| \leqslant C'$ (almost surely). Hence, by using Azuma-Hoeffding inequality, for all $\varepsilon$:

$$\mathbf{P}\left(\frac{1}{M}\sum_{m=1}^M Z_m \geqslant C(\alpha^2 + \frac{1}{\alpha K}) + \varepsilon\right) \leqslant e^{-M\varepsilon^2/C}.$$

---

[5]This is illustrated by Figure 6.

By setting $K = 1/\alpha^3$, $\varepsilon = C\alpha^2$, and taking the limit as $M$ goes to infinity, this implies that:

$$\lim_{M \to \infty} \mathbf{P} \left( \frac{1}{M} \sum_{m=1}^{M} Z_m \geqslant 3C\alpha^2 \right) = 0.$$

By using that $\frac{1}{n} \sum_{k=1}^{n} Y_k = \frac{1}{\overline{M}} \sum_{m=1}^{\overline{M}} Z_m + O(1/M)$, with $\overline{M} = \lfloor M/K \rfloor$, it follows that for all $\in \{1 \ldots d\}$:

$$\lim_{n \to \infty} \mathbf{P} \left( (\bar{\theta}_n - (\theta^* + \alpha V))_i \geqslant 3C\alpha^2 \right) = 0.$$

A symmetric proof shows that $\lim_{n \to \infty} \mathbf{P} \left( (\bar{\theta}_n - (\theta^* + \alpha V))_i \leqslant -3C\alpha^2 \right) = 0$. The theorem then follows by using a union bound on all $i$.

This shows that the constant of Theorem 3 can be chosen as $C'' = 3C$ where $C$ is the constant of Theorem 1. By choosing more carefully the values of $K$ and $\varepsilon$, this constant can be reduced to $C'' = (1 + \delta)C$ for all $\delta > 0$.

# C    Technical lemmas

This section contains the technical lemmas that are used in the proof of the main theorems. This section is divided in two parts. We first show in Lemma 6 that the derivatives of $\varphi_n$ are exponentially small as $n$ goes to infinity. Using this exponential convergence, we then prove Lemma 7 that provides bound for an *almost telescopic* sum that appears in the proof of Lemma 8. The latter shows how to bound the difference between $f(\theta_n, X_n)$ and $\bar{f}(\theta_n)$ by using a Poisson approach.

## C.1    Exponential decay of the derivatives of $\varphi_n$

The first lemma is at the core of our analysis. It shows that if the ODE has a unique attractor (as imposed by (A4)), then for $\alpha$ small enough, the derivatives of the discrete-time recurrence $\varphi_n$ are exponentially small as $n$ goes to infinity.

> **Lemma 6.** *Assume (A2)-(A4). There exist $\alpha_0 > 0$ and constants $c_1, c_2, c_3, c_4 > 0$ such that for all $\alpha < \alpha_0$, $i \in \{1, 2, 3\}$ and $n$,*
>
> *(i)* $\sum_{n=0}^{\infty} e^{-c_2 \alpha n} \leqslant c_3/\alpha$.
>
> *(ii) For all $h \in \mathcal{C}_{\leqslant 1}^i(\Theta, \mathbb{R})$, the $i$th derivative of $h \circ \varphi_n$ satisfies:*
>
> $$\left\| D^i(h \circ \varphi_n) \right\| \leqslant c_1 e^{-c_2 \alpha n},$$

*Proof.* The proof of *(ii)* is just a comparison between a sum and an integral. Indeed, as $x \mapsto e^{-x}$ is decreasing, one have:

$$\frac{1}{c_2} = \int_0^\infty e^{-c_2 x} dx = \sum_{k=1}^{\infty} \int_{(k-1)\alpha}^{k\alpha} e^{-c_2 x} dx \geqslant \sum_{k=1}^{\infty} \alpha e^{-c_2 \alpha k}.$$

This shows that:

$$\sum_{k=0}^{\infty} e^{-c_2 \alpha k} = 1 + \sum_{k=1}^{\infty} e^{-c_2 \alpha k} = 1 + \frac{1}{c_2 \alpha} = (\alpha + \frac{1}{c_2})\alpha.$$

This concludes the proof by setting $c_3 = (\alpha_0 + 1/c_2)$.

To prove *(ii)*, let us first prove the result for the first derivative ($i = 1$). By Assumption (A4), all trajectories of the ODE $\dot{\vartheta} = \bar{\vartheta}$ converge to the fixed point $\theta^*$ and the Jacobian $H := D\bar{f}(\theta^*)$ is Hurwitz. By classical results [7] on the discretization of ODE[6], this implies that, for $\alpha$ small enough,

---

[6]To see why this is true, one should remark that, because $H$ is Hurwitz, there exists $c > 0$ and $\alpha * 0$ such that for all $\alpha \leqslant \alpha_0$, the eigenvalues of $I + \alpha H$ have a modulus smaller than $1 - \alpha c$. This implies that the recurrence equation $\theta_{n+1} = \theta_n + \alpha \bar{f}(\theta_n)$ is exponentially stable around $\theta^*$. The uniform bound for all $\theta \in \Theta$ comes from the fact that $\Theta$ is compact.

the discretization $\varphi_n(\theta)$ converges exponentially fast to $\theta^*$, that is: there exists $\alpha_0 > 0$ and $a, b$ such that for all $\alpha < \alpha_0$ and $\theta \in \Theta$: $\|\varphi_n(\theta) - \theta^*\| \leqslant ae^{-c_2\alpha n}$. Using this and the fact that $\bar{f}$ is twice continuously differentiable in $\theta^*$ implies the existence of $a'$ such that:

$$\left\| D\bar{f}(\varphi_n(\theta)) - H \right\| \leqslant a'e^{-c_2\alpha n} \text{ and } (I + \alpha H)^k \leqslant a'e^{-c_2\alpha n}. \tag{14}$$

Recall that the definition of $\varphi_n$ in Equation (8) implies that $\varphi_{n+1}(\theta) - \varphi_n(\theta) = \alpha\bar{f}(\varphi_n(\theta))$ with $\varphi_0(\theta) = \theta$. Hence, as $\bar{f}$ is differentiable with respect to its initial condition, by using the chain rule, the derivative of $\varphi_{n+1}$ exists and satisfies:

$$D\varphi_{n+1}(\theta) - D\varphi_n(\theta) = \alpha D\varphi_n(\theta)D\bar{f}(\varphi_n(\theta)). \tag{15}$$

To simplify notation, let use denote by $X_{n+1} := D\varphi_{n+1}(\theta)$ (the above equation is $X_{n+1} - X_n = \alpha X_n D\bar{f}(\varphi_n(\theta))$), and let us compute[7] $X_{n+1}(I + \alpha H)^{-(n+1)}$. By adding a substracting the term $X_n(I + \alpha H)^{-n}$, it holds that:

$$\begin{aligned}
X_{n+1}(I + \alpha H)^{-(n+1)} &= X_{n+1}(I + \alpha H)^{-(n+1)} - X_n(I + \alpha H)^{-n} + X_n(I + \alpha H)^{-n} \\
&= (X_{n+1} - X_n(I + \alpha H))(I + \alpha H)^{-(n+1)} + X_n(I + \alpha H)^{-n} \\
&= \alpha X_n(D\bar{f}(\varphi_n(\theta)) - H)(I + \alpha H)^{-(n+1)} + X_n(I + \alpha H)^{-n},
\end{aligned}$$

where we used (15) for the last line.

By using that the last term $X_n(I + \alpha H)^{-n}$ corresponds to the $X_{n+1}(I + \alpha H)^{-(n+1)}$ where we replace $n + 1$ by $n$, a direct induction shows that

$$X_{n+1}(I + \alpha H)^{-(n+1)} = \alpha \sum_{k=1}^{n} X_k(D\bar{f}(\varphi_k(\theta)) - H)(I + \alpha H)^{-(k+1)}.$$

Multiplying this equation by $(I + \alpha H)^{n+1}$ implies that:

$$X_{n+1} = \alpha \sum_{k=1}^{n} X_k(D\bar{f}(\varphi_k(\theta)) - H)(I + \alpha H)^{n-k}. \tag{16}$$

By using (15), and denoting $x_{n+1} := \|X_{n+1}\|$, it holds that

$$x_{n+1} \leqslant \alpha \sum_{k=1}^{n} x_k a'e^{-c_2\alpha k}a'e^{-c_2\alpha(n-k)} = \alpha(a')^2 e^{-c_2\alpha n} \sum_{k=1}^{n} x_k. \tag{17}$$

To conclude, let $y_{n+1} := \alpha(a')^2 \sum_{k=1}^{n} y_k e^{-c_2\alpha k}$ with $y_0 := x_0$. It can be shown by induction on $k$ that $x_n \leqslant e^{c_2\alpha k}y_k$. Moreover, we have:

$$y_{n+1} = \alpha(a')^2 y_n e^{-c_2\alpha n} + y_n = y_n(1 + \alpha(a')^2 e^{-c_2\alpha n}) = \prod_{k=1}^{n}(1 + \alpha(a')^2 e^{-c_2\alpha k}).$$

By using the concavity of the log, we have that:

$$\begin{aligned}
\log \prod_{k=1}^{n}(1 + \alpha(a')^2 e^{-c_2\alpha k}) &= \sum_{k=1}^{n} \log(1 + \alpha(a')^2 e^{-c_2\alpha k}) \\
&\leqslant \sum_{k=1}^{n} \alpha(a')^2 e^{-c_2\alpha k}.
\end{aligned}$$

By using the point (i), the last quantity is bounded by some quantity $d$. This shows that $x_n \leqslant de^{-c_2\alpha n}$.

The proof of the higher derivatives (case $i \geqslant 2$) are very similar to the proof for the first derivative $i = 1$ and we only give an overview. For the second derivative, differentiating (15) with respect to $\theta$ gives:

$$D^2\varphi_{n+1}(\theta) - D^2\varphi_n(\theta) = \alpha D^2\varphi_n(\theta)D\bar{f}(\varphi_n(\theta)) + \alpha(D\varphi_n(\theta), D\varphi_n(\theta)) \cdot D^2\bar{f}(\varphi_n(\theta)).$$

---

[7]We compute this quantity to solve (15) by using a technique similar to the change of variable method.

This equation is very close to (15) with an additional term $Z_n := \alpha(D\varphi_n(\theta), D\varphi_n(\theta)) \cdot D^2\bar{f}(\varphi_n(\theta))$. Indeed, denoting by $X'_n = D^2\varphi_n(\theta)$, one can reapply the steps used to obtain (16) and get instead:

$$X'_{n+1} = \alpha \sum_{k=1}^{n} (X'_k(D\bar{f}(\varphi_k(\theta)) - H) + Z_k)(I + \alpha H)^{n-k}.$$

One can then use our result for the first derivative to show that $\left\|(X'_k(D\bar{f}(\varphi_k(\theta)) - H) + Z_k)\right\| \leqslant a''e^{-c_2\alpha k}$ and obtain an equation similar to (17) but for $x'_{n+1} := \left\|D^2\varphi(\theta)\right\|$. The proof can be modified mutatis-mutandis for higher derivatives. $\qquad\square$

## C.2  Bound on the almost-telescopic sum

In this section, we prove a lemma that we will later use in the proof of Lemma 8. This lemma concerns a summation that is *almost-telescopic*: the first (18) is of the form $\sum_{k=0}^{n-1} u_k(v_{k+1} - v_k)$. If it would hold that $u_k = u_{k-1}$, then the sum would be telescopic and there would be no need for a lemma. Here, we show that the difference $\|u_k - u_{k-1}\|$ is small and use this to prove our result. In the second lemma, we make a second round of summation.

---

**Lemma 7.** *Assume (A1)–(A4). Then, there exists a constant $C > 0$ and $\alpha_0$ such that for all $n$, $\alpha \leqslant \alpha_0$ and all $i \in \{1, 2, 3\}$, $h \in \mathcal{C}^i_{\leqslant 1}(\Theta, \mathbb{R})$ and $g \in \mathcal{C}^0_{\leqslant 1}(\Theta, \mathbb{R})$:*

$$\left| \mathbb{E}\left[ \sum_{k=0}^{n-1} D^i(h \circ \varphi_{n-(k+1)})(\theta_k) \left(g(\theta_{k+1}, X_{k+1}) - g(\theta_k, X_k)\right) \right] \right| \leqslant C. \tag{18}$$

---

*Proof.* Let us denote $u_k := D(h \circ \varphi_{n-(k+1)})(\theta_k)$ and $v_k := g(\theta_k, X_k)$. The quantity (18) is equal to $\mathbb{E}\left[\sum_{k=0}^{n-1} u_k(v_{k+1} - v_k)\right]$. By shifting the indices of the sum (which corresponds to a discrete-time integration by part), we get:

$$\sum_{k=0}^{n-1} u_k(v_{k+1} - v_k) = \sum_{k=0}^{n-1} u_k v_{k+1} - \sum_{k=0}^{n-1} u_k v_k$$

$$= \sum_{k=1}^{n} u_{k-1} v_k - \sum_{k=0}^{n-1} u_k v_k \tag{19}$$

$$= u_{n-1} v_n - u_0 v_0 + \sum_{k=1}^{n-1} (u_{k-1} - u_k) v_k. \tag{20}$$

As the norm of $g$ and $h$ are bounded by 1, $\|u_{n-1}v_n - u_0 v_0\|$ and $\|v_k\|$ are bounded independently of $\alpha$. Hence, the result follows if we can show that $\sum_{k=1}^{n-1} \|u_{k-1} - u_k\|$ is bounded regardless of $\alpha$. To show this, by adding and subtracting the term $D(h \circ \varphi_{n-k})(\theta_k)$, we have:

$$u_{k-1} - u_k = D(h \circ \varphi_{n-k})(\theta_{k-1}) - D(h \circ \varphi_{n-k})(\theta_k)$$
$$+ D(h \circ \varphi_{n-k})(\theta_k) - D(h \circ \varphi_{n-(k+1)})(\theta_k)$$

By Lemma 6, the function $D(h \circ \varphi_{n-k})$ is Lipschitz continuous with a constant $ce^{-(n-k)\alpha c_2}$. As the difference between $\theta_k$ and $\theta_{k-1}$ is of order $\alpha$, this shows that the first line is bounded by $c'\alpha e^{-(n-k)\alpha c_2}$. For the second line, the definition of the function $\varphi_{n-k}$ in (8) implies that for any $\theta$:

$$\varphi_{n-k}(\theta) - \varphi_{n-(k+1)}(\theta) = \alpha\bar{f}(\varphi_{n-(k+1)}(\theta)).$$

This shows that difference between $D(h \circ \varphi_{n-k})$ and $D(h \circ \varphi_{n-(k+1)})$ is of order $c''\alpha e^{-(n-k)\alpha c_2}$. The two facts combined imply that:

$$\sum_{k=0}^{n} \|u_{k-1} - u_k\| \leqslant (c' + c'') \sum_{k=0}^{n} \alpha e^{-c_2(n-k)\alpha} \leqslant (c' + c'') \sum_{k=0}^{\infty} \alpha e^{-c_2 k\alpha} \leqslant (c' + c'')c_3\alpha.$$

$\qquad\square$

## C.3 Poisson equation and treatment of the averaging term

For a given $g : \Theta \times \mathcal{X} \to \mathbb{R}$, we say that the function $G_g : \Theta \times \mathcal{X} \to \mathbb{R}$ is a solution of the Poisson equation if it satisfies

$$\sum_{y \in \mathcal{X}} K_{x,y}(\theta)(G_g(\theta, x) - G_g(\theta, y)) = g(\theta, x) - \bar{g}(\theta), \tag{21}$$

with $\bar{g}(\theta) = \sum_x \pi_x(\theta) g(\theta, x)$ as before.

It is shown in [1, Lemma 4] that under assumption (A3), for each $g$, there exists such a function $G_g$. Moreover, under the regularity assumption (A2), this same lemma shows that the function $G_g$ can be chosen to be four times differentiable and that is derivatives satisfy $\left\| D^{\leqslant i} G_g \right\| = \mathcal{O}(\left\| D^{\leqslant i} g \right\|)$.

---

**Lemma 8.** *Assume (A1)–(A4). Then, there exists a constant $C > 0$ and $\alpha_0$ such that for all $n$, $\alpha \leqslant \alpha_0$ and all $i \in \{1, 2, 3\}$, $h \in \mathcal{C}^i_{\leqslant 1}(\Theta, \mathbb{R})$ and $g \in \mathcal{C}^0_{\leqslant 1}(\Theta, \mathbb{R})$:*

$$\left| \sum_{k=0}^{n-1} \mathbb{E} \left[ D^i(h \circ \varphi_{n-(k+1)})(\theta_k) \Big( g(\theta_k, X_k) - \bar{g}(\theta_k) \Big) \right] \right| \leqslant C. \tag{22}$$

---

*Proof.* To lighten the notations, we consider the case $i = 1$, the proof holds for $i > 1$ by replacing $D(h \circ \varphi)$ by $D^i(h \circ \varphi)$. By adding and subtracting some terms, the quantity (22) is equal to the expectation of

$$\sum_{k=0}^{n-1} D(h \circ \varphi_{n-k-1})(\theta_k) \Big( g(\theta_k, X_k) - \bar{g}(\theta_k) + G_g(\theta_k, X_k) - G_g(\theta_k, X_{k+1}) \tag{23}$$

$$+ G_g(\theta_k, X_{k+1}) - G_g(\theta_{k+1}, X_{k+1}) \tag{24}$$

$$+ G_g(\theta_{k+1}, X_{k+1}) - G_g(\theta_k, X_k) \Big). \tag{25}$$

We examine the three lines separately:

- By using the fact that $X_{k+1}$ makes a Markovian transition (2), and by using the definition of $G_h(x, y)$, we get:

$$\mathbb{E}\left[ G_g(\theta_k, X_k) - G_g(\theta_k, X_{k+1}) \mid \mathcal{F}_k \right] = -\sum_{y'} K(\theta_k)_{X_k, y'} G_g(\theta_k, y')$$

$$= -(g(\theta_k, X_k) - \bar{g}(\theta_k))$$

  from which follows that the expectation of (23) is equal to zero.

- To bound Equation (24), we use the definition (1) of $\theta_{k+1}$ to implies that there exists a constant $C > 0$ such that $|\theta_{k+1} - \theta_k| \leqslant C\alpha$. By using that $G_g$ is Lipschitz-continuous (see [1, Lemma 4]) and that the norm of the derivative of $h \circ \varphi_{n-k}$ is bounded by $c_1 e^{-c_2(n-k)\alpha}$ (Lemma 6), this shows that there exists $c' < \infty$ such that:

$$\mathbb{E}\left[ \sum_{k=0}^{n-1} \| D(h \circ \varphi_{n-k-1})(\theta_k) \left( G_g(\theta_k, X_{k+1}) - G_g(\theta_{k+1}, X_{k+1}) \right) \| \right] \leqslant c'\alpha \sum_{k=0}^{\infty} e^{-c_2 k\alpha} \leqslant c_3 c'.$$

- The fact that the expectation of (25) is bounded is a direct consequence of Lemma 7 applied with $g = G_g$ and the fact that $\| DG_g \| = \mathcal{O}(\| Dg \|)$.

This conclude the proof. $\qquad \square$

## C.4 Refined treatment of the averaging term

The next two lemmas prove properties of the average sums that refine the analysis of Lemma 8. They are used to study the term $\overline{A}_n$ in the proof of Proposition 5.

> **Lemma 9.** *Assume (A1)–(A4). Then, there exists a constant $C > 0$ and $\alpha_0$ such that for all $n$, $\alpha \leqslant \alpha_0$:*
>
> $$\forall \ell \in \mathcal{C}^1_{\leqslant 1}(\Theta, \mathbb{R}) : \qquad \frac{1}{n}\sum_{k=0}^{n-1} \mathbb{E}\left[\ell(\theta_k, X_k) - \bar{\ell}(\theta_k)\right] \leqslant C(\frac{1}{n} + \alpha). \qquad (26)$$
>
> $$\forall \ell \in \mathcal{C}^2_{\leqslant 1}(\Theta, \mathbb{R}) : \qquad \frac{1}{n}\sum_{k=0}^{n-1} \mathbb{E}\left[\ell(\theta_k, X_k) - \bar{\ell}(\theta_k) + \alpha\bar{m}(\theta_k)\right] \leqslant C(\frac{1}{n} + \alpha^2), \qquad (27)$$
>
> *where $\bar{m} = \sum_{x,x' \in \mathcal{X}} DG_\ell(\theta, x')(f(\theta, x) + R(\theta, x, x'))K_{x,x'}(\theta)\pi_x(\theta)$.*

*Proof.* By applying the same trick as for (23)–(25) in the one of Lemma 8 but replacing $D^i(h \circ \varphi_{n-(k+1)})(\theta_k)$ by $1/n$, the left-hand-side of (26) is equal to the expectation of

$$\frac{1}{n}\sum_{k=0}^{n-1} \underbrace{G_\ell(\theta_k, X_{k+1}) - G_\ell(\theta_{k+1}, X_{k+1})}_{=\,\mathcal{O}(\alpha \|DG_\ell\|) \text{ because } DG_\ell \text{ is bounded.}} + \frac{1}{n}\sum_{k=0}^{n-1} \underbrace{G_\ell(\theta_{k+1}, X_{k+1}) - G_\ell(\theta_k, X_k)}_{=\,O(1) \text{ because sum is telescopic.}} \qquad (28)$$

where there is no equivalent of (23) because we already use that the expectation of this term is 0.

In (28), each element of the first sum is $\mathcal{O}(\alpha)$ by using a Taylor expansion of $G_\ell$ and using that $\|DDG_\ell\| = \mathcal{O}(\|D\ell\|)$. Hence the first term is of order $\mathcal{O}(\alpha)$. The second sum is telescopic. As it is multiplied by $1/n$, this leads to the term $\mathcal{O}(1/n)$ of (26). Combining both leads to (26).

To obtain (27), we refine the analysis of the first term of (28). By using a Taylor expansion on the function $\theta \mapsto G_\ell(\theta, X)$, the quantity $G_\ell(\theta_k, X_{k+1}) - G_\ell(\theta_{k+1}, X_{k+1})$ is equal to

$$DG_\ell(\theta_k, X_{k+1})(\theta_k - \theta_{k+1}) + \left\|D^{\leqslant 2}G_\ell\right\| \|\theta_{k+1} - \theta_k\|^2. \qquad (29)$$

By definition, $\|\theta_{k+1} - \theta_k\|^2$ is of order $\alpha^2$.

Let $m(\theta, x) = \sum_{x' \in \mathcal{X}} DG_\ell(\theta, x')(f(\theta, x) + R(\theta, x, x'))K_{x,x'}(\theta)$. By using the law of total expectation,

$$\begin{aligned}
\mathbb{E}\left[DG_\ell(\theta_k, X_{k+1})(\theta_k - \theta_{k+1})\right] &= -\alpha\mathbb{E}\left[DG_\ell(\theta_k, X_{k+1})(f(\theta_k, X_k) + M_{k+1})\right] \\
&= -\alpha\mathbb{E}\left[\mathbb{E}\left[\mathbb{E}\left[DG_\ell(\theta_k, X_{k+1})(f(\theta_k, X_k) + M_{k+1}) \mid \mathcal{F}_k \wedge X_{k+1}\right] \mid \mathcal{F}_k\right]\right] \\
&= -\alpha\mathbb{E}\left[m(\theta_k, X_k)\right].
\end{aligned}$$

By using (26) with the function $m$ instead of $\ell$, we obtain that $\frac{1}{n}\sum_{k=0}^{n-1} \mathbb{E}\left[m(\theta_k, X_k) - \bar{m}(\theta_k)\right] \leqslant C(\alpha + 1/n)$. This shows that the first term of (28) is equal to $\alpha \sum_{k=0}^{n-1} \bar{m}(\theta_k)$ plus a term of order $C(\alpha^2 + \alpha/n)$. As the term $\mathcal{O}(1/n)$ from the telescopic sum of the second term of (28) dominates the term $\alpha/n$, this implies (27). $\qquad\square$

> **Lemma 10.** *Assume (A1)–(A4). Then, there exists a constant $C > 0$ and $\alpha_0$ such that for all $n$, $\alpha \leqslant \alpha_0$, $h \in \mathcal{C}^4_{\leqslant 1}(\Theta, \mathbb{R})$ and $g \in \mathcal{C}^3_{\leqslant 1}(\Theta, \mathbb{R})$:*
>
> $$\left|\frac{1}{N}\sum_{n=1}^{N}\sum_{k=0}^{n-1} D(h \circ \varphi_{n-(k+1)})(\theta_k)(g(\theta_k, X_k) - \bar{g}(\theta_k)) - \sum_{k=0}^{N-1} \bar{m}(\theta_k)\right| \leqslant C\left(\alpha + \frac{1}{N\alpha^2}\right),$$
>
> *where the function $\bar{m}$ is equal to*
>
> $$\bar{m}(\theta) = \sum_{x,x' \in \mathcal{X}}\sum_{n=0}^{\infty} D\big(D(h \circ \phi_n)(\theta)G_g(\theta, x')\big)\big(f(\theta, x) + R(\theta, x, x')\big)K_{x,x'}(\theta)\pi_x(\theta).$$

*Proof.* Let us denote by $u_{n-(k+1)}(\theta_k) := D(h \circ \varphi_{n-(k+1)})(\theta_k)$ and let us study the sum $S := \frac{1}{N}\sum_{n=1}^{N}\sum_{k=0}^{n-1} D(h \circ \varphi_{n-(k+1)})(\theta_k)$. By reordering the sum, this is equal to:

$$S = \frac{1}{N}\sum_{k=0}^{N-1}\sum_{n=k+1}^{N} u_{n-(k+1)}(\theta_k)g(\theta_k, X_k) = \frac{1}{N}\sum_{k=0}^{N-1}\sum_{n=0}^{N-(k+1)} u_n(\theta_k)g(\theta_k, X_k).$$

$$= \frac{1}{N}\sum_{k=0}^{N-1}\sum_{n=0}^{\infty} u_n(\theta_k)g(\theta_k, X_k) - \frac{1}{N}\sum_{k=0}^{N-1}\sum_{n=N-k}^{\infty} u_n(\theta_k)g(\theta_k, X_k) \quad (30)$$

By Lemma 6, there exists $c_1, c_2 > 0$ such that $\|u_n\| \leqslant c_1 e^{-c_2\alpha n}$. This shows that the second term of the previous sum (30) is bounded (in absolute value) by

$$\frac{1}{N}\sum_{k=0}^{N-1}\frac{c}{\alpha}e^{-c_2\alpha(N-k)} \leqslant \frac{c'}{N\alpha^2}, \quad (31)$$

for a suitable constant $c'$. This term goes to 0 when $N$ goes to infinity.

To treat the first term of (30), we apply Lemma 9 with the function $\ell(\theta, X) := \sum_{n=0}^{\infty} D^i(h \circ \varphi_n)(\theta)g(\theta, X) = \sum_{n=0}^{\infty} u_n(\theta)g(\theta, X)$. $\qquad \square$

### C.5 Characterization of $V$

The next Lemma show how an approximate computable expression of the bias term $V$ can be obtained. Recall that by definition, $V$ is dependent on $\varphi$ with step-size $\alpha$. The subsequently defined and computable expression $\tilde{V}$ is resolves this dependence on $\alpha$ and is further justified as it admits an accurate of order $\alpha$ with respect to $V$.

---

**Lemma 11** (Approximate Computable Expression of $V$). *Assume (A1)–(A4). Define*

$$V = Dh(\theta^*)\big(H^{(1)}\big)^{-1}(S + H^{(2)} \cdot W) + D^2h(\theta^*) \cdot W \quad (32)$$

*with $H^{(1)}_{i,j} := \frac{\partial \bar{f}_i}{\partial \theta_i}(\theta^*)$ is the Jacobien matrix of $\bar{f}$ at $\theta^*$, $H^{(2)}_{i,jk} = \frac{\partial \bar{f}_i}{\partial \theta_i \partial \theta_j}(\theta^*)$ is the second derivative of $\bar{f}$ in $\theta^*$. In the above notation, $H^{(2)} \cdot W := (\sum_{j,k} H^{(2)}_{i,jk} W_{j,k})_{i=1...d}$, where $W$ is the unique solution to the Sylvester equation $H^{(1)}W + W\big(H^{(1)}\big)^T + O = 0$ and where $O$ and $S$ are defined by*

$$O := \sum_{x,x'} G_f(\theta^*, x')\big(f(\theta^*, x) + R(\theta^*, x, x')\big)^T K(\theta^*)_{x,x'}\pi_x(\theta^*) + \bar{g}(\theta^*), \quad (33)$$

$$S := \sum_{x,x'} \big(f(\theta^*, x) + R(\theta^*, x, x')\big)DG_f(\theta^*, x')^T K(\theta^*)_{x,x'}\pi_x(\theta^*) \quad (34)$$

*with $\bar{g}(\theta^*) := \sum_x \pi_x(\theta^*)\Big((f(\theta^*, x) - \bar{f}(\theta^*))^{\otimes 2} + Q(\theta^*, x)\Big)$ is defined as in the proof of Proposition 5.*

*There exists a constant $C > 0$ and $\alpha_0$ such that for all $\alpha \leqslant \alpha_0$:*

$$\Big|V - V^{(\alpha)}\Big| \leqslant C\alpha,$$

*where $V^{(\alpha)} = \alpha(\sum_{k=0}^{\infty} D^2(h \circ \varphi_k)(\theta^*)\bar{g}(\theta^*) + m(\theta^*))$ is defined as in the proof of Proposition 5.*

---

Note that the constant $V$ does not depend on $n$ or $\alpha$.

*Proof.* Using the definitions (33) and (34) for $O$ and $S$ respectively, and the definition of $V$, $\bar{g}$ and $m$ in the proof of Proposition 5, we can rewrite $V^{(\alpha)}$ as:

$$V^{(\alpha)} = \alpha\Big(\sum_{k=0}^{\infty} D(h \circ \varphi_k)(\theta^*)S + \sum_{k=0}^{\infty} D^2(h \circ \varphi_k)(\theta^*) \cdot O\Big). \quad (35)$$

Using that $\varphi_k(\theta^*) = \theta^*$ yields the identity

$$V^{(\alpha)} = \alpha Dh(\theta^*) \sum_{k=0}^{\infty} D\varphi_k(\theta^*)S \tag{36}$$

$$+ \alpha Dh(\theta^*) \sum_{k=0}^{\infty} D^2\varphi_k(\theta^*) \cdot O + \alpha D^2 h(\theta^*) \cdot \sum_{k=0}^{\infty} D\varphi_k(\theta^*)OD\big(\varphi_k(\theta^*)\big)^T, \tag{37}$$

where the first line corresponds to the first term of (35) and the second line to the second term of (35). The notations correspond to $Dh(\theta^*)D^2\varphi_k(\theta^*) = (\sum_i \frac{\partial h}{\partial \theta_i} \frac{\partial \varphi_i}{\partial \theta_m \partial \theta_n})_{m,n}$ and $\cdot$ denotes the sum over the element wise product between the matrices.

From hereon out, we suppress the dependence of the therms on $\theta^*$ in order to ease the notation. As said before, it is shown in [1, Lemma 4], under assumptions (A2) and (A3), that $G_f$ is computable and four times differentiable, which implies that the terms $S$ and $O$ are computable. Therefore, in the rest of the proof we are concerned with obtaining computable expressions for the terms

$$\alpha \sum_{k=0}^{\infty} D\varphi_k \ ; \qquad \alpha \sum_{k=0}^{\infty} D\varphi_k OD\varphi_k^T \ ; \qquad \alpha \sum_{k=0}^{\infty} D^2\varphi_k \cdot O. \tag{38}$$

We recall that by definition $D\varphi_1 = I + \alpha H^{(1)}$. Using the chain rule, we have $D\varphi_k = D(\varphi_1 \circ \varphi_{k-1}) = (I + \alpha H^{(1)})D\varphi_{k-1} = (I + \alpha H^{(1)})^k$. By assumption of exponential stability, $H^{(1)}$ is Hurwitz and thus, for small enough $\alpha$, all its eigenvalues have negative real parts. This implies that for such $\alpha$, $\sum_{k=0}^{\infty}(I + \alpha H^{(1)})^k = (\alpha H^{(1)})^{-1}$. This gives the first term $Dh(\theta^*)\big(H^{(1)}\big)^{-1}S$ of (32).

Define $W^{(\alpha)}$ as the second sum of (38) which is equal to

$$W^{(\alpha)} := \alpha \sum_{k=0}^{\infty} D\varphi_k OD\varphi_k^T$$

$$= \alpha \sum_{k=0}^{\infty}(I + \alpha H^{(1)})^k O(I + \alpha(H^{(1)})^T)^k.$$

As $H^{(1)}$ is Hurwitz, $W^{(\alpha)}$ is well-defined for $\alpha$ small enough and is a solution to the discrete-time Sylvester equation $(I + \alpha H^{(1)})W^{(\alpha)}(I + \alpha H^{(1)})^T - W^{(\alpha)} + \alpha O = 0$. To obtain a computable expression independent of $\alpha$, we consider the corresponding continuous-time Sylvester equation given by $H^{(1)}W - W(H^{(1)})^T + O = 0$ for which the Hurwitz property of $H^{(1)}$ ensures the existence of a unique solution $W$. By definition of the two Sylvester equations, we then have that $W^{(\alpha)} = W + \alpha H^{(1)}W(H^{(1)})^T$, i.e., the difference between the two solutions is of order $\alpha$. This gives the second term $Dh(\theta^*)\big(H^{(1)}\big)^{-1}H^{(2)} \cdot W$ of (32).

For the last sum of (38), apply the chain rule and substitute $D\varphi_1 = I + \alpha H^{(1)}$ and $D^2\varphi_1 = \alpha H^{(2)}$ to obtain the identity

$$D^2\varphi_k = \left(\frac{\partial \varphi_k}{\partial \theta_m \partial \theta_n}\right)_{i,mn}$$

$$= \left(\sum_{j=1}^{k} \sum_a \big((I + \alpha H^{(1)})^{k-j}\big)_{i,a} \sum_{b,c} \alpha H_{a,bc}^{(2)} \big((I + \alpha H^{(1)})^j\big)_{b,m} \big((I + \alpha H^{(1)})^j\big)_{c,n}\right)_{i,mn}.$$

Therefore,

$$D^2\varphi_k \cdot O = (\sum_{j,k} \frac{\partial \varphi_k}{\partial \theta_j \partial \theta_k} O_{j,k})_{i=1...d}$$

$$= \sum_{j=1}^{k}(I + \alpha H^{(1)})^{k-j} \alpha H^{(2)} \cdot (I + \alpha H^{(1)})^j O(I + \alpha(H^{(1)})^T)^j.$$

with $H^{(2)} \cdot (I + \alpha H^{(1)})^j O(I + \alpha(H^{(1)})^T)^j = \sum_{m,n} H^{(2)}_{i,mn} \left((I + \alpha H^{(1)})^j O(I + \alpha(H^{(1)})^T)^j\right)_{m,n}$.
Consequently, for $\sum_{k=0}^{\infty} D^2\varphi_k \cdot O$ the above yields

$$\alpha \sum_{k=0}^{\infty} \sum_{j=1}^{k} (I + \alpha H^{(1)})^{k-j} \alpha H^{(2)} \cdot (I + \alpha H^{(1)})^j O(I + \alpha(H^{(1)})^T)^j$$

$$= \alpha \sum_{j=1}^{\infty} \sum_{k=j}^{\infty} (I + \alpha H^{(1)})^{k-j} \alpha H^{(2)} \cdot (I + \alpha H^{(1)})^j O(I + \alpha(H^{(1)})^T)^j$$

$$= \alpha \sum_{j=1}^{\infty} \sum_{k=0}^{\infty} (I + \alpha H^{(1)})^k \alpha H^{(2)} \cdot (I + \alpha H^{(1)})^j O(I + \alpha(H^{(1)})^T)^j$$

$$= \alpha \sum_{j=1}^{\infty} (\alpha H^{(1)})^{-1} \alpha H^{(2)} \cdot (I + \alpha H^{(1)})^j O(I + \alpha(H^{(1)})^T)^j$$

$$= (\alpha H^{(1)})^{-1} \alpha H^{(2)} \cdot (I + \alpha H^{(1)}) \left(\alpha \sum_{j=0}^{\infty} (I + \alpha H^{(1)})^j O(I + \alpha(H^{(1)})^T)^j\right)(I + \alpha(H^{(1)})^T)$$

$$= (\alpha H^{(1)})^{-1} \alpha H^{(2)} \cdot (I + \alpha H^{(1)}) W^{(\alpha)} (I + \alpha(H^{(1)})^T)$$

$$= (H^{(1)})^{-1} H^{(2)} \cdot W^{(\alpha)} + \alpha C',$$

with $C' = \left((H^{(1)})^{-1} H^{(2)} \cdot \left(H^{(1)} W^{(\alpha)} + W^{(\alpha)}(H^{(1)})^T + \alpha H^{(1)} W^{(\alpha)}(H^{(1)})^T\right)\right)$ which is bounded by assumption. Using the previously discussed difference between $W$ and $W^{(\alpha)}$, we conclude that $(H^{(1)})^{-1} H^{(2)} \cdot W$ is an approximate solution to $\sum_{k=0}^{\infty} D^2\varphi_k \cdot O$ with error of order $\alpha$. $\qquad\square$

