# OpenReview forum: "Computing the Bias of Constant-step Stochastic Approximation with Markovian Noise"
_NeurIPS.cc/2024/Conference — NeurIPS 2024 poster_

### Official Review · Reviewer_Hzq1 · 2024-07-08

**Soundness:** 3
**Presentation:** 3
**Contribution:** 3
**Rating:** 5
**Confidence:** 4

**Summary:**

This paper studies stochastic approximation with constant stepsize and Markovian noise. The authors provide a characterization of the bias (i.e., the difference between the expectation of the iterate and the desired limit), show that Polyak-Ruppert averaging can help reduce the variance but not the bias, and numerically demonstrate that Richardson-Romberg extrapolation helps reduce the bias.

**Strengths:**

The writing is mostly clear and the proof is written with high quality. The result is an extension of [17,36] to the case where the Markov chain can have its transition probability matrix as a function of the stochastic iterate, which enables the authors to apply the results to RL algorithms with time-varying behavior policies (though the results are not formally presented in this paper). Overall, this is a strong theoretical work and the techniques developed in this work are quite novel.

**Weaknesses:**

(1) Assumption (A2) is relatively strong and difficult to verify in practice.

(2) The authors claim that Assumption (A4) implies the global exponential stability of the ODE, but did not provide a reference for the claim. Also, the claim in lines 466-471 needs formal justification.

**Questions:**

Both theorems require $h$ to be uniformly bounded by a constant. This is not true even for $h(\theta)=\theta$, which corresponds to bounding the expectation of $\theta_n$. Am I missing something?

**Limitations:**

The results are mostly asymptotic. A characterization of the stochastic approximation algorithm behavior for a finite $n$ would be preferred for practical purposes.

---

> ### Author Response · Authors · 2024-08-01
> **h(theta)=theta is ok**
>
> In the question, the reviewer asks if using h(theta)=theta is ok because this function is not bounded. This function can indeed be used because we assume that theta remains bounded. Note that one could also replace this assumption by a weaker assumption on the moments (see the answer to rev 2).

---

> ### Comment · Reviewer_Hzq1 · 2024-08-11
> **Is there a rebuttal?**
>
> I want to confirm if the authors have uploaded their rebuttal. I am not seeing it here.

---

> > ### Author Response · Authors · 2024-08-11
> >
> > The rebuttal was kept very short and should be visible to all. There are answers to specific comments below each review (they were not visible to all before but they should be now).

---

### Official Review · Reviewer_Mbpd · 2024-07-11

**Soundness:** 3
**Presentation:** 2
**Contribution:** 3
**Rating:** 7
**Confidence:** 4

**Summary:**

This paper studies the asymptotic bias in non-linear stochastic approximation algorithms with Markovian noise and fixed step-size. Upon applying the averaging technique of Polyak and Ruppert, the authors identify that, in general, the bias is of the same order of the step-size. The main source of bias is characterized, and an extrapolation technique is employed so that bias is attenuated. Finally, a few numerical studies are presented for illustration of the theoretical contributions.

**Strengths:**

I find the overall contribution of the paper to be very interesting. As far as the reviewer is aware, the characterization of bias for nonlinear SA with parameter dependent Markovian noise is novel and original.

The analysis seems sound. The reviewer did not have time to look over every single proof carefully, but haven't found egregious errors in the proofs revised.

The problem setup, assumptions and approach to analysis are clearly stated. The paper is well-written, but some polishing is needed.

**Weaknesses:**

In particular, there have been recent papers, achieving similar bias characterizations and higher order error bounds for stochastic approximation with Markovian noise (see for example [R1] and [R2] which both deal with linear recursions). The approach to analysis in [R2] is similar to the present paper in the sense that the bias characterization is also given in terms of solutions to Poisson's equations. Moreover, this is not the first time that Richardson-Romberg extrapolation is used for bias attenuation in stochastic approximation: it was previously proposed in [R1] to kill the dominant term of order O(\alpha) just like the present paper.

The authors do cite [R1] in the present work, but the reviewer feels that a deeper discussion is needed on how the present paper improves upon [R1] and [R2]. I encourage the authors to include a citation on previous uses of this technique as well.


[R1] Huo, Dongyan, Yudong Chen, and Qiaomin Xie. "Bias and extrapolation in Markovian linear stochastic approximation with constant stepsizes." Abstract Proceedings of the 2023 ACM SIGMETRICS International Conference on Measurement and Modeling of Computer Systems. 2023.

[R2] Lauand, Caio Kalil, and Sean Meyn. "The curse of memory in stochastic approximation." 2023 62nd IEEE Conference on Decision and Control (CDC). IEEE, 2023.

**Questions:**

- Is it possible to relax the assumption that the parameters remain within the compact set \Theta with probability one?   It seems likely that the authors can obtain their results subject to a moment bound, which I believe is more realistic based upon standard SA stability theory.

- One thing that sets this paper apart from others is that they allow Markovian noise that is parameter dependent.   This is extremely valuable in RL applications such as Q-learning (for example, epsilon-greedy policies induce such a model), and actor-critic methods.  Could the authors provide more discussion regarding applications in which the Markovian noise is parameter dependent?

-Could the authors clarify whether the experiments display in Figure 1 (a) and (b) pertain to the same run? It is hard to identify the improvement between RR exploration and the regular algorithm for the choice of \alpha =0.01. I encourage the authors to include sample paths for \alpha =0.01 and \alpha =0.005 instead of the \alpha =0.0025 for visualization.

**Limitations:**

The authors discuss the limitations of their work and assumptions in section 3.3.

---

> ### Author Response · Authors · 2024-08-01
> **answers to comments**
>
> - about bounded theta: the reviewer is correct by suggesting that we can replace the assumption of "bounded theta" by an assumption that would control the probability that theta is "far" from theta^*. For instance, a bound on the higher moment of theta would work.
>
> - about the markov noise being theta-dependent: thank you for your positive comments. We will discuss these motivating examples if the final version.
>
> - Figure 1: yes, the runs of the two figures are the same (this is why the red curve is duplicated on the right panel, in order to have some comparison). Lower values of alpha tend to provide slower convergence rate, which is why they are not shown there.

---

> > ### Comment · Reviewer_Mbpd · 2024-08-11
> >
> > I would like to thank the authors for their responses.
> >
> > **Bounded theta** Given that a moment bound on theta could replace the current assumption that theta lives in a compact set, I encourage the authors to modify that in the final version ( or at least provide some discussion on that) so that the assumptions of the paper are more realistic.
> >
> > **Figure 1** I understand that a lower value of $\alpha$ might lead slower convergence, but I believe that the current plot is not helpful in showing the improvement between RR exploration and the regular algorithm. In encourage the authors to use a larger value of $\alpha$ so that a plot with $\alpha^2$ (without RR exploration) could be displayed and compared to the extrapolated estimates without much worry on convergence rates.

---

> > > ### Author Response · Authors · 2024-08-12
> > >
> > > Thank you for your comments. Our plan for the funak version is to:
> > > - add a formal statement about unbounded theta.
> > > - add more numerical experiments as suggested.

---

### Official Review · Reviewer_xFdD · 2024-07-17

**Soundness:** 2
**Presentation:** 2
**Contribution:** 2
**Rating:** 5
**Confidence:** 3

**Summary:**

The paper studies non-linear stochastic approximation scheme ($\theta_n$) driven by a uniformly geometrically ergodic MC $(X_n)$. Moreover, it is allowed the evolution of the Markovian noise $X_n$ to depend on $\theta_n$.  The authors study the asymptotic behaviour of the last iterate, Polyak-Ruppert and Richardson-Romberg procedures for differentiable test functions. The result generalizes recent result of Aymeric Dieuleveut, Alain Durmus, and Francis Bach. Bridging the gap between constant step size stochastic gradient descent and markov chains. The Annals of Statistics, 48(3):pp. 1348–1382, 2020.

**Strengths:**

Novel technique based on infinitesimal generator comparison

**Weaknesses:**

- It would be good to provide non-asymptotic results rather than asymptotic
- I think that the result of theorem 3 is very weak since it is obtained using Chebyshev’s inequality. Could you please comment on $\alpha^{5/4}$ term?

**Questions:**

- I think it is better to give a definition of a unichain.
- Is it possible to relax conditions of the theorems? For example, what will be if we reduce the number of derivatives in Th 2? Is it true that in this case one will obtain \alpha^{3/2} in the remainder term? Is it possible to achieve this using the suggested technique?

**Limitations:**

-

---

> ### Author Response · Authors · 2024-08-01
> **answer on hiw to obtain more general results**
>
> Dear reviewer,
>
> Thank you for your detailed ans positive review. Here are some answers to your comments / questions:
>
> - yes, the bound of alpha^(3/4) of Theorem is probably not optimal. This Theorem could in fact be a called a corrolary of Theorem 2. Using a more advanced concentration inequality would probably require a bound on the exponential moment which are not direct from Theorem 2.
>
> - unchain just means that the Markov component has a unique stationary for all parameter theta. We will precise that.
>
> - the derivation of the bias term in alpha requires to have twice differentiable functions. Our proof could be adapted to show that if the functions are only twice differentiable, then the reminder term is a o(alpha). The alpha^(3/4) for 3 times differentiable functions is not direct from our analysis.
>
> - about non asymptotic-results: some of our propositions are in fact non asymptotic but the main results are stated jn an asymptotically way to be cleaner.

---

> > ### Comment · Reviewer_xFdD · 2024-08-12
> >
> > I thank the authors for their response. I retain my current score.

---

### Author Rebuttal · Authors · 2024-08-01

The authors would like to thank all reviewers for their detailed and constructive reviews. All reviews are correct and suggest interesting improvements that will be invluded in the final version.

Most of the comments or questions are answered below each review.

---

### Decision · Program_Chairs · 2024-09-25

**Decision:**

Accept (poster)

**Comment:**

This paper considers stochastic approximation (SA) under Markovian noise and constant step-size, which is commonly employed in practice. Authors rely on
the generator of the process to study the bias of SA algorithm, and derive its order
under smoothness conditions. Authors also provide bias orders for the Polyak averaged iterates, and extensions involving Richardson-Romberg extrapolation.


This paper was reviewed by three expert reviewers w/ the following Scores/Confidence: 5/3, 7/4, 5/4. I think the paper is studying an interesting topic and the results are relevant to NeurIPS community. The following concerns were brought up by the reviewers:

- The main results in this paper are asymptotic. The non-asymptotic versions should be clearly stated in discussion/conclusion section as future work.

- Better discussion connecting this work to the existing literature is needed.

- Authors should discuss how restrictive their conditions are.


Authors should carefully go over reviewers' suggestions and address any remaining concerns in their final revision. Based on the reviewers' suggestion, as well as my own assessment of the paper, I recommend including this paper to the NeurIPS 2024 program.